# Analyzing land suitability for key cereal crops in Mansa Watershed, Southwest Ethiopia

**Wude Taye, Wakshum Shiferaw** ⓘ *, **Genaye Tsegaye**

Arba Minch University, College of Agricultural Sciences, Natural Resource Management, Arba Minch, Ethiopia

* waaqsh@yahoo.com

## Abstract

This study is vital for helping farmers to produce appropriate crops based on physical land suitability and for assisting land use planners in decision-making. Large-scale crop production is essential to supply raw materials for industries, focusing on areas with high production potential to boost yields and meet growing demand. However, physical land suitability analysis for major cereal crops is lacking in Mansa Watershed of Southwest Ethiopia. Therefore, this research aimed to assess the physical land suitability for key crops such as wheat, maize, and teff. Utilizing the FAO land evaluation framework, the study employed various data sets, including Sentinel-2A satellite images, soil data, climate information, and elevation models, to determine suitability factors. The Analytical Hierarchy Process (AHP) was used for pairwise comparison of parameters, while Geographic Information System (GIS) software's weighted overlay tool was applied to evaluate suitability for the specified crops. A vector overlay was utilized for land allocation for each crop. The analysis considered ten criteria: soil pH, depth, texture, drainage, organic matter, slope, altitude, rainfall, temperature, and land use change. Results indicated that approximately 29.6%, 61%, and 50% of the study area were moderately suitable for maize, teff, and wheat production, respectively. Additionally, 52.8%, 38.8%, and 13.9% of the area were marginally suitable for these crops, while 17.6% and 36% of the area were unsuitable for maize and wheat, respectively. Overall, 44% of the land was moderately suitable, and 10% was marginally suitable for the selected crops. Notably, there were no areas classified as highly suitable; most lands were identified as moderately or marginally suitable. Moving forward, sustainable land management practices are necessary to enhance land suitability and improve soil health. Further analyses should also consider irrigation facilities, market access, and processing industries to provide more options for stakeholders and policymakers.

**Data availability statement:** All relevant data are within the paper.

**Funding:** The author(s) received no specific funding for this work.

**Competing interests:** The authors has no conflict of interest.

## Introduction

Evaluation of land suitability is crucial for improving food security, both globally and specifically in Ethiopia. Land suitability refers to the fitness of a specific type of land for a particular use. As one of the most important natural resources, land plays a vital role in supporting livelihoods and meeting the increasing demands for food, fiber, feed, and fuel [1]. This is especially true for countries whose economies are primarily based on agriculture. Therefore, land suitability assessment is essential for sustainable land use planning and management [2]. Land suitability assessment can be a key tool for identifying resource potentials and environmental constraints that can provide alternatives for better sustainable land use planning [3]. The physical suitability of land focuses on factors such as climate, land morphology, and soil quality [4]. FAO [5] recommends assessing land suitability for crop production based on climatic, topographical, and soil characteristics. Analyzing land suitability is a valuable tool for planning and managing land resources [6].

Ethiopia faces significant challenges in food security with a growing population and heavy reliance on natural resources [7]. Ethiopia has a rapidly growing population of over 100 million [8], making it one of the fastest-growing populations in Africa. This puts immense pressure on land resources resulted in land degradation and loss of arable land. Therefore, ensuring suitable land for agricultural production is crucial to mitigate the risk of food insecurity and sustain the growing population [9]. In this regard, land suitability assessment is the base for sustainable land resources planning and management [10].

Ethiopia cultivates a variety of crops due to diverse agroecologies, favorable climate conditions, and fertile soils [11]. Cereals like teff, wheat, maize, sorghum, and barley are major staple crops and contribute to food security [12]. However, factors such as inappropriate land use, overgrazing, deforestation, and climate change have contributed to environmental degradation and reduced agricultural outputs. The study area, Mansa watershed, allows for the cultivation of various grains, but poor land use and soil nutrient depletion have led to insufficient crop yields [13].

Adopting land suitability evaluation is essential for sustainable land resource management for instance, through reducing soil erosion to boost agricultural production [14]. Land suitability analysis combines multiple factors to make informed decisions [15]. Analytic hierarchical process (AHP) is one of the most widely used MCDM techniques. This technique is frequently used in conjunction with geographic information systems (GIS) to evaluate ecological capabilities in land suitability and natural resource management, as well as to establish the relative weight of decision criteria [16]. Remote sensing (RS) and geographic information systems (GIS) are examples of machine learning tools that can be effectively combined to create intelligent systems for land use planning [17]. Results from machine learning (ML) techniques are more accurate because they create a new model using statistical analysis and computational algorithms [18]. The operation and representation of geospatial data in suitability analysis has been done in recent years using geographic information systems [19]. These technologies can be used efficiently in systems involving information, coordination, control, and communication. A multitude of outputs, such as

maps of land use and cover, obstacles, slope, road mobility, and line-of-sight plots, can be produced using satellite remote sensing data [20].

We identified potentially hostile environments and selected better locations for border control and military strategy using GIS, machine learning, and RS-based techniques [21]. These methods can also yield more comprehensive data and information, which can be used as a foundation for choosing a site. One well-known application using geospatial data is site selection through the use of GIS, RS, analytical hierarchy process, and machine learning [22]. The use of machine learning, the analytical hierarchy method, and GIS are frequently used tools that let users choose Decision-making is aided through the integration of multicriteria evaluation techniques with geographic information systems, which foster the management and organization of vast volumes of geographic data [23]. While multicriteria techniques evaluate alternatives based on the subjective values and priorities of the decision-maker, GIS performs deterministic overlay and buffer operations in site selection problems [24]. Decision-makers can select the most suitable location with the aid of multicriteria decision systems and GIS [25]. GIS and MCE together provide a potent tool for selecting key suitable crop sites and generating excellent analytical outputs.

Furthermore, a computer program used Genetic Algorithm for Rule Set Production (GARP) uses genetic algorithms to build ecological niche models for various species such as cereal crops. The environmental parameters such as heat, precipitation, and elevation, etc., are the species should be able to sustain population levels are described by the generated models. Local species observations and associated environmental parameters are used as input, describing possible boundaries of the species' capacity for survival. Geographic information systems frequently store such types of environmental parameters. A random collection of mathematical formulas, also known as limiting environmental conditions, makes up a GARP model. Every rule is used as randomly assembled to produce a limitless amount of models that could potentially describe the possibility of the species [26].

The integration of various software including ArcMap, ERDAS imagine, and IDRISI using the Sentinel-2A satellite images. Multi-Criteria Decision Making (MCDM) with Analytical Hierarchy Process (AHP) matrix are calculated which provides decision-makers with relevant maps and location databases to consider arable land for better agricultural production. However, MCDM and AHP are not new algorithms but methods. It has been used in many spatial modeling studies. A branch of operations research known as multiple-criteria decision-making (MCDM) or multiple-criteria decision analysis (MCDA) formally assesses several competing criteria when making decisions. According to Gharye et al. [27], it is also referred to as multi-objective decision analysis, multiple attribute utility theory, multiple attribute value theory, and multiple attribute preference theory. Additionally, the Analytic Hierarchy Process (AHP), a popular MCDM technique that facilitates decision-making, enables decision-makers to rank options based on a set of criteria. An analytical method for classifying and assessing complex decisions is the Analytic Hierarchy Process (AHP). Saaty developed AHP in the 1970s to help decision-makers evaluate and prioritize options based on a set of standards [27]. This study aimed to conduct a physical land suitability analysis using GIS, RS, and integrated multi-criteria to identify suitable areas for teff, wheat, and maize cultivation in Mansa watershed. Factors such as soil, climate, land use/ land cover, and topography were considered. The study seeks to identify the physical factors that determine the suitability of these crops, evaluate the land suitability, and allocate suitable land for each crop within the watershed. The findings of this study will provide valuable insights for decision-makers in sustainable land use planning and agricultural production.

## Materials and methods

### Description of the study area

This study was conducted in Mansa watershed, situated in Dawuro zone of Southwest Ethiopia National Regional State. Geographically, the study area lies between 6°42′ 30″N – 7°4′30″N and 36°53′0″E–37°15′0″E at an altitude ranging from 607 m to 2763 m and having total area of 1024.23 km² [28].

**Climate:** The mean annual temperature varied from 15.02°C to 21.31°C and the average annual rainfall varied between 1529–1663 mm (Fig 1). The study area is characterized by a bimodal rainfall pattern. The short rainy season was between February and March and the long rainy season was between June and September [28].

**Topography and soil:** The Mansa watershed is characterized by diverse topographical features, including valleys, plains, hills, and mountains. The study area comprises six soil types, namely dystric nitisols, dystric fluvisols, dystric gleysols, orthic acrisols, leptosols, and eutric cambisols. Among these, dystric nitisol is the most dominant followed by cambisol and orthic-acrisol [19]. The understanding of soil information is crucial for land use planning.

Demographic and crop production: According to annual statistics from the BoFED [29], the watershed has a population of approximately 73,158, with 37,409 males and 35,749 females [28], with a significant reliance on subsistence agriculture and livestock farming. Dominant crops include maize, teff, and wheat, alongside other produce like barley, enset, and various fruits.

### Research methods

**Research design and approach.** This research employed a quantitative design, utilizing tables and graphs to represent data from factors such as DEM, rainfall, slope, soil, and temperature, enabling the assessment of suitable land for selected cereal crops.

**Data source and software used.** This study utilized both spatial and non-spatial data from various organizations (Table 1).

Various software types, including ArcMap 10.3, ERDAS IMAGINE 2015, IDRISI 17.0, and Google Earth, were employed for a range of activities in the creation of land suitability maps (Table 2).

**Parameters used for land suitability analysis of cereal crops.**

### a. Crop requirement and criteria rating for land suitability

This study utilized the FAO [5] framework for land suitability analysis, evaluating environmental criteria for maize, teff, and wheat based on climate, topography, and soil characteristics. The criteria were developed from FAO guidelines and existing literature on land suitability for small-scale rain-fed agriculture [5,30–37], summarized in Table 3.

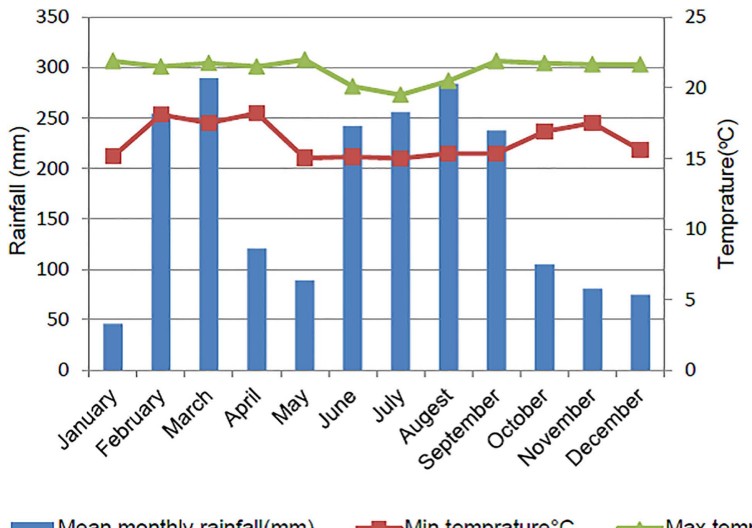

**Fig 1. Rainfall and temperature distribution of the study area.**

## b. Standardization and weighting the criteria

The criteria vector maps were converted to raster data formats for standardization. Linear scale transformation was used to standardize the factors, following the FAO standard for land evaluation for rain-fed agriculture [36]. The factors were categorized into four categories namely highly suitable, moderately suitable, marginally suitable, and not suitable using value ranging from 1 to 4. The weighted factor was determined using the pairwise comparison matrix and AHP matrix with values ranging from 1 to 9 assigned to each factor [37]. The weighting was based on literature reviews, the real nature of the study area, and guidelines from [5]. The purpose of weighting was to express the importance of each factor relative to others in crop yield and growth rate. The priority weights were calculated using the pairwise comparison matrix and eigenvector values.

$$\text{eigen vector} = \text{Aji} = \frac{\sum_{i=1}^{n} \left( \frac{w1}{w1} * \frac{w1}{w2} * \ldots * \frac{w1}{wn} \right)^{\frac{1}{n}}}{\Sigma \left[ \sum_{i=1}^{n} \left( \frac{w1}{w1} * \frac{w1}{w2} * \ldots * \frac{w1}{wn} \right)^{\frac{1}{n}} \right]} \quad (1)$$

Where w1 is the sum of rows for pairwise comparison and n is the size of the matrix

The consistency ratio (CR) was calculated to verify the consistency of comparison as (Table 4).

$$\text{CI} = \frac{(\lambda max - n)}{n - 1} \quad (2)$$

Where CI is the consistency index, n is the number of elements being compared in the matrix, and λmax is the largest or principal eigenvalue of the matrix.

$$\text{CR} = \frac{CI}{RI} \quad (3)$$

**Table 1. Summary of data sources and their purpose.**

| Data set | Format of data | Scale/Resolution | Source of data | Functions |
|---|---|---|---|---|
| Digital Elevation Model (DEM) | Tiff | 30x30m | https://earthexplorer.usgs | To generate elevation and slope map of the study area |
| Climate data | MS Excel | | Ethiopia National Meteorology Agency (NMA) | To interpolate temperature and rainfall data |
| Sentinel-2A image of 2021 | Tiff | 10m*10m | https://earthexplorer.usgs.gov | To develop a land use land cover map of the study area |
| Soil data | Shape File | at a scale of 1:250,000 | Agricultural Transformation Agency (ATA) | To develop soil OM, pH, Depth, Drainage and Texture map of the study area |

**Table 2. Software types and their purposes.**

| Software used | Purpose |
|---|---|
| ERDAS IMAGIN 2015 | Image processing, data analysis and LULC classification |
| ArcGIS 10.3 | Data processing, calculating, classifying, overlay analysis and map preparation |
| IDRISI Selva 17.0 | For weight derivation of factors |
| Google earth | Used for visual interpretation tool for accuracy assessment for LULC classification |

**Table 3. Environmental requirements for maize, teff, and wheat under rain-fed agriculture.**

| Crops | Factor | Unit | Range of Suitability | | | |
|---|---|---|---|---|---|---|
| | | | S1 | S2 | S3 | N |
| Teff | Rainfall | Mm | 450–550 | 300–450/ 550–800 | 800–1200 | <200 >1200 |
| | Temperature | ºc | 15–21 | 14–15/21–22 | 12–14/ 22–23 | <12/>25 |
| | Elevation | M | 1600–2200 | 1000–1600/ 2200–2400 | 2400–2800 | <1000 >2800 |
| | Slope | % | 0–7 | 7–15 | 15–25 | >25 |
| | Soil PH | – | 5.5–7.5 | 5.2–5.5 and 7.5–7.8 | 5.0–5.2 and 7.8–8.0 | <4.5 >8.5 |
| | Soil OM | % | >3 | 2–3 | 1–2 | <1 |
| | Soil depth | Cm | >50 | 30–50 | 20–30 | <10 |
| | Soil texture | Class | C,Si,SiC, | SiC, | siL.Cl,SL | L,SCL, |
| | Soil drainage | Class | W, MM | I | SE, E | P, VP |
| | LULU | Type | CL | Grassland | BShL | WL,BL |
| Wheat | Rainfall | Mm | 450–650 | 350–450, 650–850 | 300–350, 850–1000 | <300 >1000 |
| | Temperature | ºc | 14.9–18.4 | 14.4–14.9 /18.4–19.4 | 13.4–14.4, 19.4–20.8 | <13 >20.8 |
| | Elevation | M | 1600–2200 | 1000–1600/ 2200–2400 | 2400–2800 | <1000 >2800 |
| | Slope | % | 0–13 | 13–25 | 25–40 | >40 |
| | Soil pH | – | 6.0–8.0 | 5.2–6.0 and 8.0–8.3 | 5.0–5.2 and 8.3–8.5 | <5 and >8.5 |
| | Soil OM | % | >3 | 2–3 | 1–2 | <1 |
| | Soil depth | Cm | >100 | 75–100 | 50–75 | <50 |
| | Soil texture | Class | C,Si,SiC,SiL,SC | L,CL.SicL | SCL | LS,S |
| | Soil drainage | Class | W,SE | MW | I | VP |
| | LULU | Type | CL | Grassland | BShL | WL,BL |
| Maize | Rainfall | Mm | 500–750 | 450–500/ 750–1200 | 300–400 1200–1600 | <300 >1600 |
| | Temperature | ºc | 15–21 | 14–15/21–22 | 12–14/ 22-23 | <11/>25 |
| | Elevation | M | 1600–2200 | 1000–1600/ 2200–2400 | 2400–2800 | <1000 >2800 |
| | Slope | % | 0–7 | 7–15 | 15–25 | >25 |
| | Soil pH | – | 5.5–7.5 | 5.2–5.5 and 7.5–7.8 | 5.0–5.2 and 7.8–8.0 | <4.5 >8.5 |
| | Soil OM | % | >3 | 2–3 | 1–2 | <1 |
| | Soil depth | Cm | >50 | 30–50 | 20–30 | <10 |
| | Soil texture | Class | SI,SiC,C | Sic | SiL,CL,SC | L,SCL,S,SL |
| | Soil drainage | Class | W | Mw | VP | E,SE |
| | LULU | Type | CL | Grassland | BShL | WL,BL |

S1 = Highly suitable, S2 = Moderately suitable, S3 = Marginally suitable, N = Not suitable, P = poorly drained, VP = very poorly drained, SP = somewhat poorly drained, MW = moderately well drained = well drained, SE = somewhat excessively drained = excessively drained, I = imperfectly, C = Clay, CL = Clay loam; SCL = Silt-clay-loam, SiL = Silty-loam, L = Loam, SL = Sandy-loam, CL = Crop Land, BShL = Bush land, WL = woody land, BL = bare land.

**Table 4. Matrix size and Random consistency index.**

| Matrix size | 1 | 2 | 3 | 4 | 5 | 6 | 7 | 8 | 9 | 10 | 11 | 12 | 13 | 14 | 15 |
|---|---|---|---|---|---|---|---|---|---|---|---|---|---|---|---|
| Random consistency index(RI) | 0 | 0 | 0.58 | 0.9 | 1.12 | 1.24 | 1 | 1.41 | 1.45 | 1.49 | 1.51 | 1.48 | 1.56 | 1.57 | 1.59 |

Random Index [27].

Where CR is the consistency ratio, CI is the consistency index, RI is the random index.

If the CR ≤ 0.10, it means the pairwise comparison matrix has a suitable consistency. Otherwise, If CR ≥ 0.10 it implies that pairwise consistency has inadequate consistency [38]. Following the standards weight, a weighted overlay technique was applied to arc map 10.3 to generate suitability maps for each land use type (Equation 4), and a vector overlay analysis was performed to create a composite suitable land allocation map:

$$s = \sum_{i=0}^{n}(WiXi)$$

(4)

Where S is the suitability, Wi is the weight of factor i and Xi is the criterion score of factor i.

**Method of data analysis.** Soil data, including depth, drainage, texture, organic matter, and pH, were compiled and imported into ArcGIS 10.3 for visualization and analysis. Rainfall and temperature data in Excel were examined for missing values and estimated using neighboring stations. Raster images were created from the input data in ArcGIS, and point data were interpolated using the Inverse Distance Weighting tool. An elevation map was derived from a 30 m resolution Digital Elevation Model (DEM), and slope calculations were performed using the Spatial Analyst tool in ArcGIS.

For land use and land cover analysis, Sentinel-2A imagery from 2021 was pre-processed with ERDAS IMAGINE 2015, where images were layer stacked and classified using both supervised and unsupervised methods. Unsupervised classification identified clusters and the number of groups, while supervised classification refined these definitions for thematic maps. The classified images were validated with 240 random points using Google Earth imagery. Satellite images were sourced from USGS Earth Explorer (https://earthexplorer.usgs.gov/), which were downloaded [38]. With image processing and interpretation for land cover mapping conducted by Rwanga and Ndambuki [39].

**Classification of images:** Both supervised and unsupervised classification techniques were used to categorize the pre-processed photos. Using user-provided training sets (signatures) and field-collected ground truth data, the maximum likelihood algorithm is used in the supervised classification technique to classify the image. For the supervised classification, 240 validation or random points (signatures) were utilized for each land use type and land cover. Consequently, the study area's land cover classes were determined. Land covers change detection analysis.

A base map of the study areas was created using Google Earth satellite imagery and ground truth data. Several features in the study area were identified by manually combining satellite images. The area coverage of the first year was subtracted from that of the second year, as indicated in Equation (1) following Islam [40], to determine the magnitude of change for each LULC class. The difference between the final year and the initial year, which indicates the magnitude of change between corresponding years, was divided by the number of study years to obtain the annual rate of change for each LULC.

**Accuracy assessment:** ArcGIS was used to evaluate the accuracy of the supervised land use classification for the 2021 image. 240 randomly generated points for 2021 supervised and unsupervised images were obtained from the classifier. When the data sets were trained during supervised land use classification, the software itself was able to identify the unique colour tone and pixel value of each and every point. These figures served as the standard values. The user then identified each point that was generated at random and assigned it to a different class. The points that were correctly identified were taken into account for classification. This reference and classified data were also used to create an error matrix and Kappa statistics. The overall accuracy was determined from the error matrix by dividing the total number of examined

pixels by the sum of the entries that form the major diagonal. The following equations were also used to calculate the Kappa coefficient of agreement [41].

**Accuracy assessment:** To assess the accuracy of the output map and determine if it meets the required standards, an accuracy assessment was conducted [42–45]. The purpose of this assessment was to evaluate quantitatively how effectively the pixels were classified into the correct land cover classes within the study area. It is recommended by Congalton [46] to have a minimum of 30 sample points per land use/land cover class for accuracy assessment. However, collecting reference samples, especially through ground surveys can be costly. In this study, a minimum of 30 samples were collected for each land use/land cover class to assess the accuracy of the land use/land cover classification. To express the classification accuracy, the most common method used is the error matrix also known as the confusion matrix as described by Lillesand et al. [47]. This matrix provides a cross-tabulation of the classified land cover and the actual land cover determined through the sample site results [45].

**Kappa coefficient:** Kappa coefficient is a measure of overall agreement of a matrix and it is the ratio of the sum of diagonal values to total number of cell counts in the matrix. It describes the proportionate reduction in error generated by the classification process compared with the error of completely random classifications. Kappa values are also characterized into 3 groups: a value greater than 0.75 (75%) represents strong agreement defined as excellent, a value between 0.4 and 0.7 (40–70%) represents moderate agreement (good classification) and a value less than 0.4 (40%) represents poor agreement [48].

$$K = \frac{N \sum_{i=1}^{r} xii - \sum_{i=1}^{r} (xi + *x + i)}{N^2 - \sum_{i=1}^{r} (xi + *xi+)}$$

Where,

K = kappa coefficient

r = number of columns and rows in error matrix,

N = total number of observations,

Xii = observation in column i and row i,

Xi+ = marginal total of row i, and

X+i = marginal total of column i.

In general, the methodology of this study can be summarized in the following conceptual framework (Fig 2).

## Ethics statement

To get data for this study, IRB or ethics committee who approved or waived the study was not formed. Rather our institution, Arba Minch University wrote a letter to woreda where the Mansa watershed exists. Then the woreda assigned experts to link the researchers with concerned bodies in the study watershed.

## Results and discussions

### Physical factors influencing the suitability of major cereal crops

The soil pH analysis conducted in this study revealed that the soil in the study area is highly acidic with pH values ranging from 3.08 to 5.33. The reclassified soil pH suitability assessment showed that only 29.50% of the watershed is moderately suitable for teff, wheat, and maize crops. About 41.30% of the watershed is marginally suitable, while 29.2% is not suitable for these crops (Table 5). The results indicate that soil pH is a limiting factor for the cultivation of the selected cereal crops in the study area, given the high acidity of the soil. This finding aligns with previous studies by Fekadu and Negese [49] who identified soil pH as a primary limiting factor in the Yikalo sub-watershed. Similarly, Girma et al. [50] found that soil pH and fertility were limiting factors for wheat and maize cultivation in the Jello Watershed. However, the area in which moderately suitable with pH value was larger area than finding by Alemayehu [51] for wheat production in Sinana Research

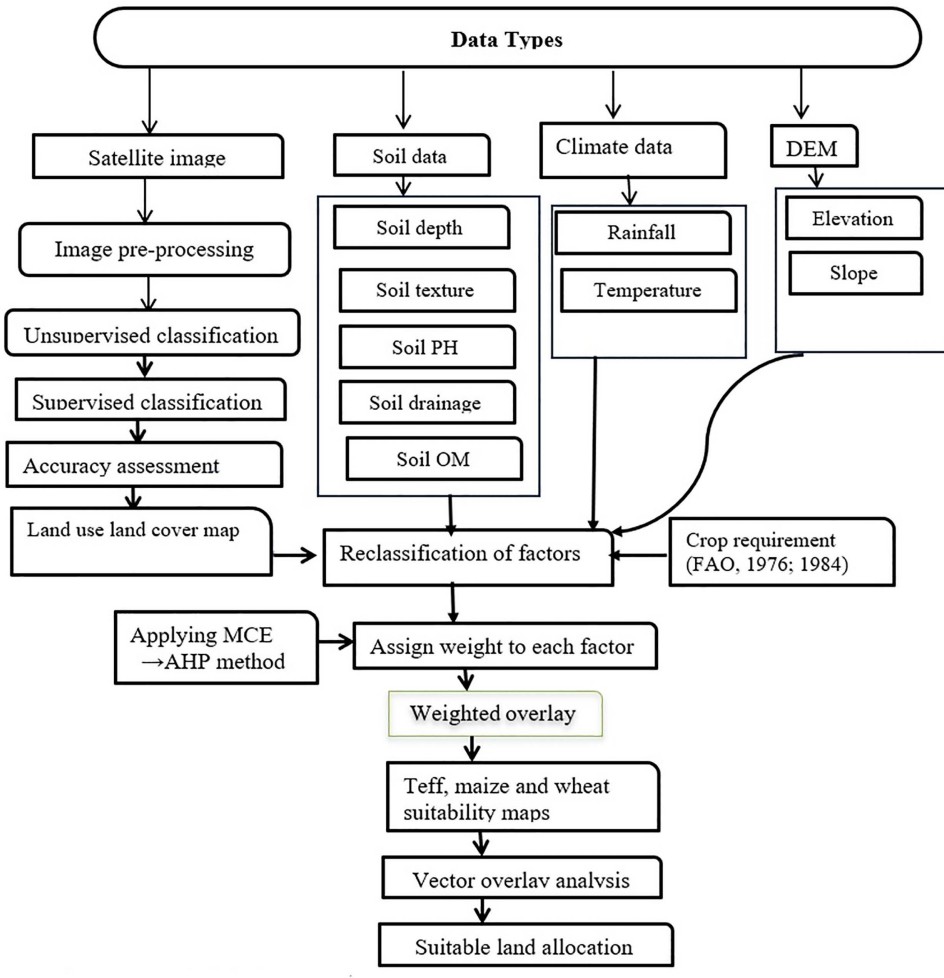

**Fig 2. Conceptual framework of the study.**

Site of Southeastern Ethiopia. Yohannes and Soromessa [42] identified texture, temperature, slope, and erosion hazard as the main limiting factors for wheat and barley production. However, Girmay et al. [52] discovered that the soil pH in the Gateno watershed has potential for wheat, barley, and faba bean crops.

The soil depth within the watershed ranges from 34 to 100 cm. The reclassified soil depth suitability analysis showed that 70.33% of the area is highly and moderately suitable for teff and maize crops. Specifically, 70.33% of the watershed has moderate suitability for wheat, and the remaining 29.67% is not suitable (Table 6). The highly and moderately suitable soil depths for teff and maize crops were located in Northwest and Northeast of Mansa Watershed, but not suitable

**Table 5. Spatial variation of soil pH.**

| Criteria | Class | Suitability level | Value | Area (km²) | Area (%) | Crop type |
|---|---|---|---|---|---|---|
| Soil pH | 5.2-5.5 | Moderately suitable | 2 | 302.15 | 29.50 | Teff, wheat and maize |
| | 5.0-5.2 | Marginally suitable | 3 | 423.01 | 41.30 | Teff, wheat and maize |
| | <4.5 | Not suitable | 4 | 299.08 | 29.20 | Teff, wheat and maize |

lands were exited in the center of the watershed (Fig 3). The analysis indicates that the study area has potential for teff and maize cultivation in terms of soil depth, but there are limitations for wheat crops. This finding aligns with the research by Yohannes and Soromessa [42], who identified soil depth as a primary limiting factor for wheat and barley cultivation in the Andit Tid watershed. Similarly, Selassie et al. [53] reported that shallow soil depth and low water holding capacity influenced maize production in the Yigossa Watershed. However, the study by Motuma et al. [54] in Wogdie Woreda found that most of the study area had a high suitability in terms of soil depth for wheat and sorghum crops. However, in this

**Table 6. Spatial variation of soil depth.**

| Criteria | Class (cm) | Suitability level | Value | Area (km²) | Area (%) | Crop type |
|---|---|---|---|---|---|---|
| Soil Depth | >50 | Highly suitable | 1 | 720.35 | 70.33 | Teff and Maize |
| | 34–50 | Moderately suitable | 2 | 303.89 | 29.67 | |
| | 75–100 | Moderately suitable | 2 | 720.35 | 70.33 | Wheat |
| | <50 | Not suitable | 4 | 303.89 | 29.67 | |

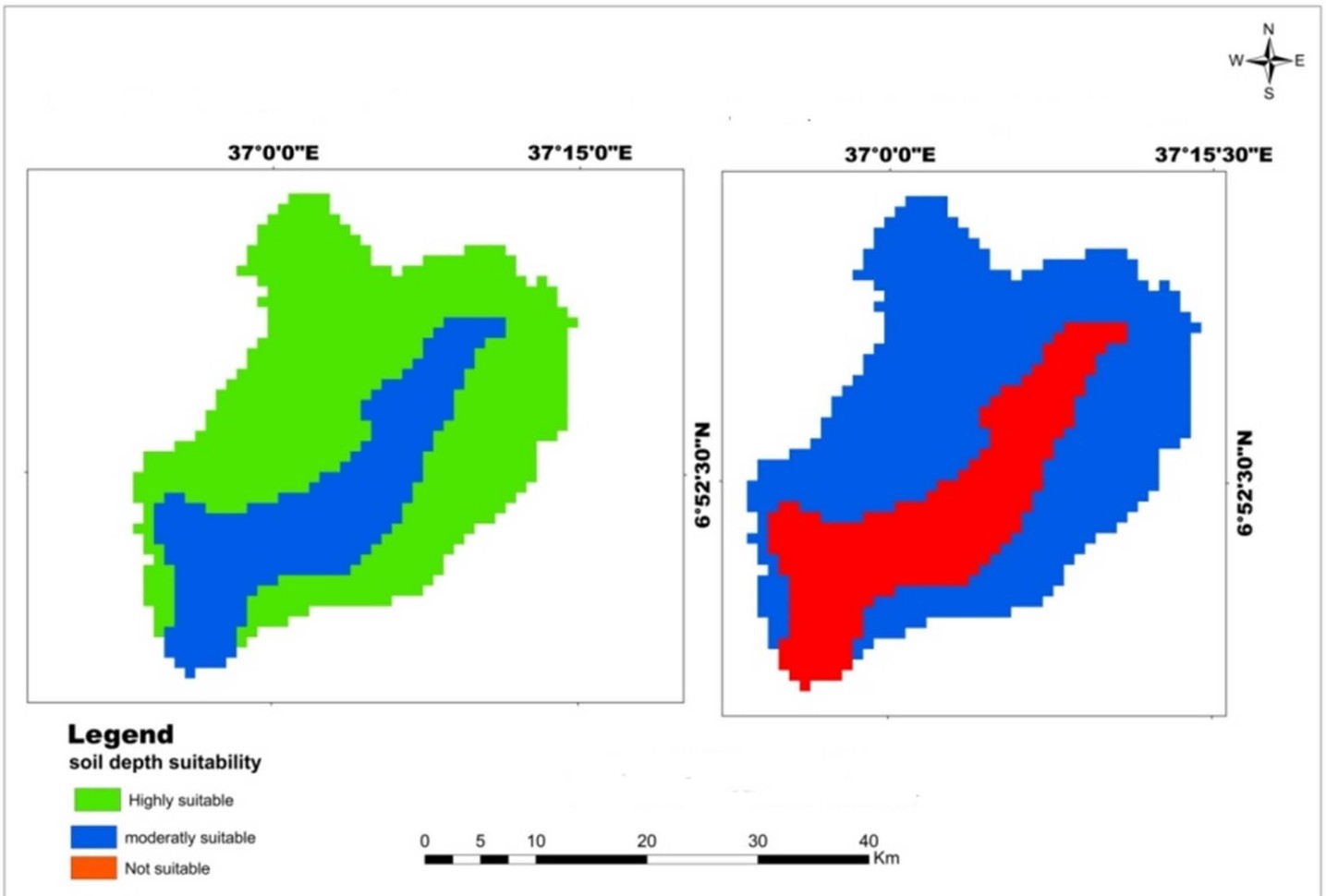

**Fig 3. Land suitability map of classified soil depth for teff, maize, and wheat.**

study, the area moderately suitable for wheat production was larger area than study by Alemayehu [51] for wheat production in Sinana Research Site of Southeastern Ethiopia.

The watershed drainage classes are as follows moderately well-drained and imperfectly drained. The watershed's reclassified soil drainage map showed that 70.33% and 29.67% of the watershed is highly and moderately suitable for teff and about 70.33% of the watershed is moderately suitable for wheat and maize and 29.67% of the watershed is marginally suitable for wheat and maize crops (Table 7). The highly and moderately suitable soil drainage for teff and maize crops were located in Northwest and Northeast of Mansa Watershed, but marginaly suitable lands were exited in the center of the watershed (Fig 4). The analysis result showed that most part of the watershed has a potential for the cultivation of each selected cereal crop. This result was more consistent with the finding of Mosisa et al. [55] showed that most part of the watershed has good potential for maize crop production. Similarly Girmay et al. [52] found that the soil drainage of Gateno watershed has

Table 7. Spatial variation of soil drainage.

| Criteria | Class | Suitability level | Value | Area (km²) | Area (%) | Crop type |
|---|---|---|---|---|---|---|
| Soil Drainage | Moderately well drainage | Highly suitable | 1 | 720.35 | 70.33 | Teff |
| | Imperfectly drainage | Moderately suitable | 2 | 303.89 | 29.67 | |
| | Moderately well drainage | Moderately suitable | 2 | 720.35 | 70.33 | Wheat and maize |
| | Imperfectly drainage | Marginally suitable | 3 | 303.89 | 29.67 | |

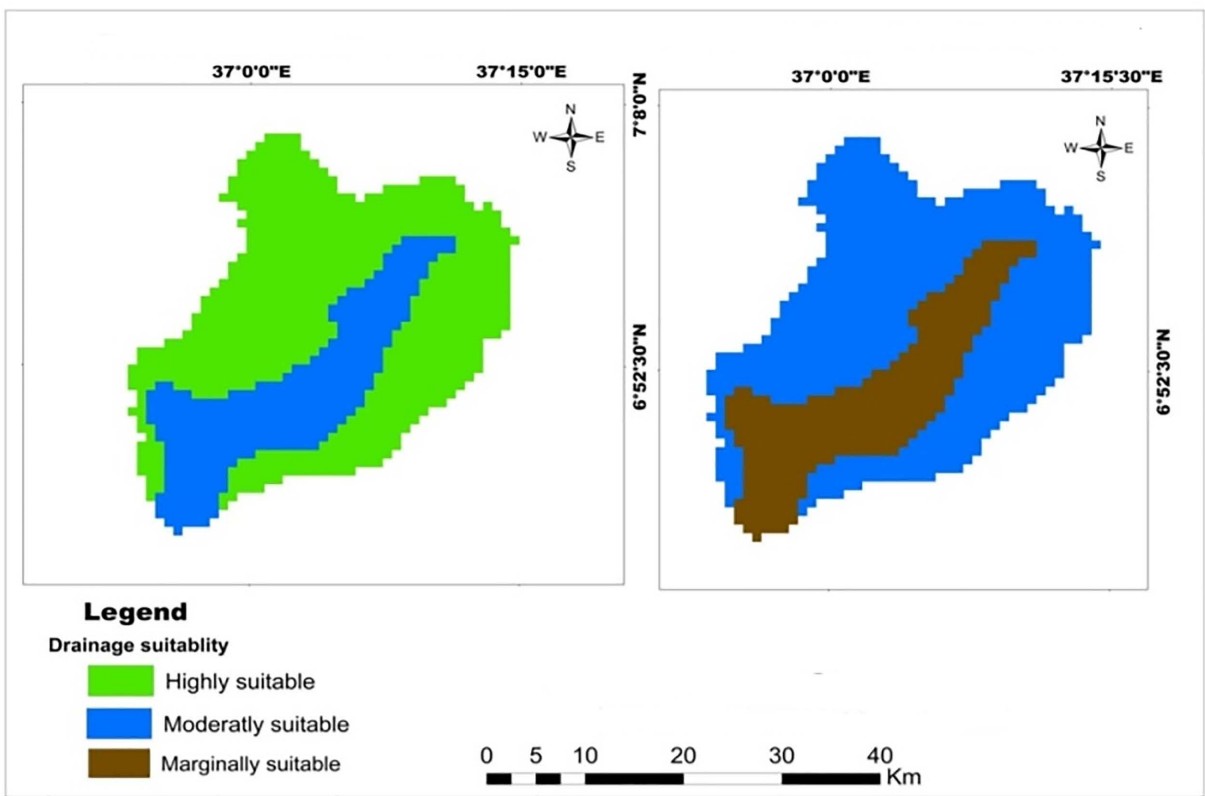

Fig 4. Land suitability map of classified soil drainage for teff, maize, and wheat.

potential for production of wheat, barley and faba bean crop. However, the area of soil in Mansa watershed had less drained for wheat production than that of area of wheat production in Sinana Research Site of Southeastern Ethiopia [51].

In Mansa watershed soil organic matter content ranges from 0.223 to 1. 3307. Results for soil organic matter showed that 70.33% and 29.67% of the watershed is marginally suitable and unsuitable for each crop, respectively (Table 8). The majority of marginally suitable soil organic matter content ranges for teff and maize crops was located in Northwest and Northeast of Mansa Watershed, but not suitable lands were exited in the center of the watershed (Fig 5). The required soil organic matter for teff, maize and wheat can be grow successfully in soils with OM content greater than 3% but the

**Table 8. Spatial variation of soil organic matter.**

| Criteria | Class | Suitability level | Value | Area (km²) | Area (%) | Crop type |
|---|---|---|---|---|---|---|
| Soil OM | 1–2 | Marginally suitable | 3 | 720.35 | 70.33 | Teff, wheat and maize |
| | <1 | Not suitable | 4 | 303.89 | 29.67 | Teff, wheat and maize |

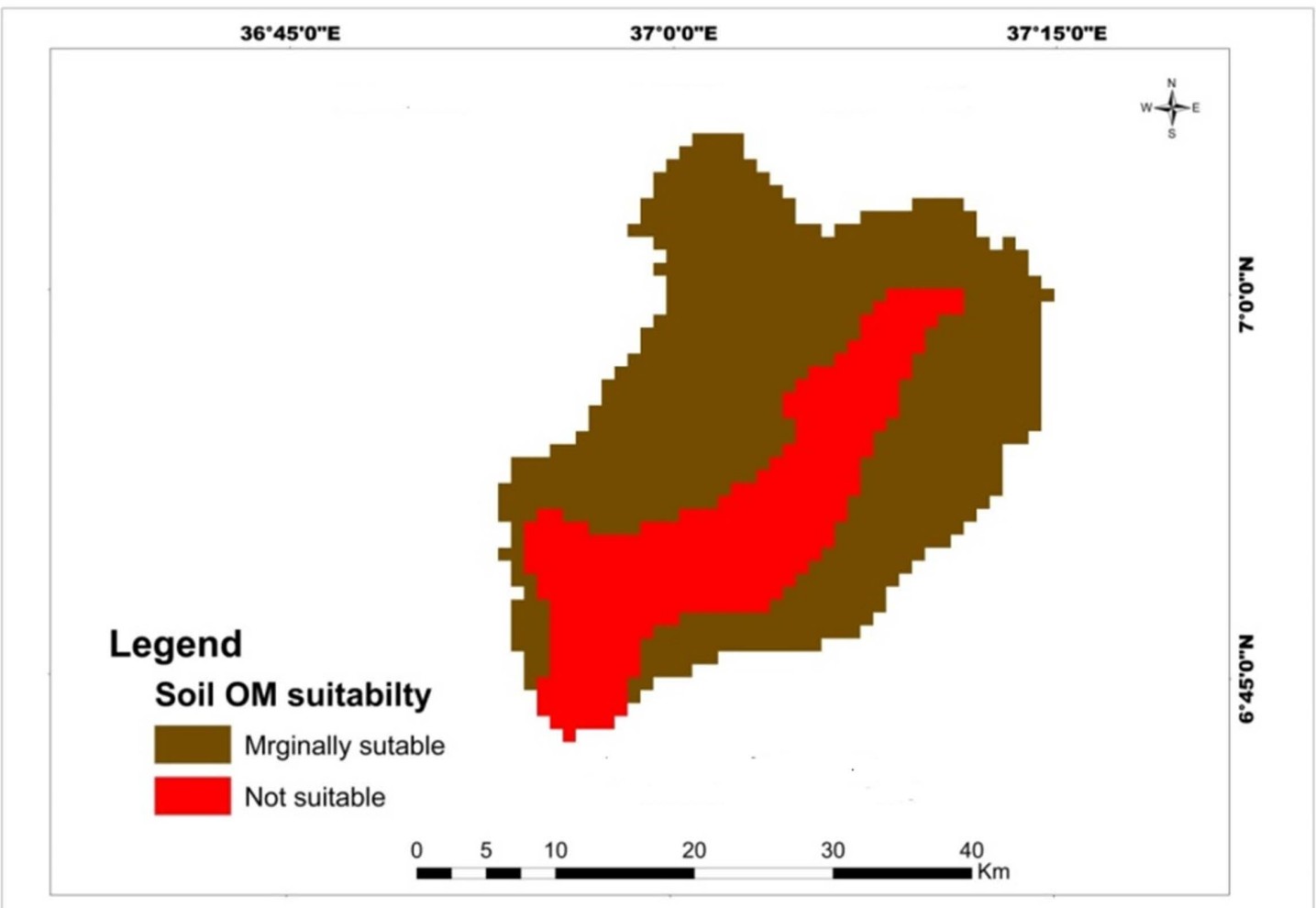

**Fig 5. Land suitability map of classified soil organic matter for teff, maize, and wheat.**

watershed soil organic matter is less than 2%. This indicates that soil organic matter is one of the determinant factors in the study area. In line with this finding, soil organic was the main limiting factor for the production of maize in Abobo of western Ethiopia [56]. Similarly, Selassie et al. [53] reported that the main suitability limitations for cereal crops were low organic matter contents and acidic soil pH in Yigossa watershed. Bahir et al. [37] also found that soil organic matter was the main limiting factors in selecting land suitability in Gerado catchment, North-Eastern Ethiopia for wheat and maize production.

The Mansa watershed is predominantly characterized by clay and clay loam soil textures. A reclassification of the soil texture data was conducted to determine the suitability of these textures for the selected cereal crops. The reclassification resulted in three classes. The reclassified soil texture map of the watershed reveals that approximately 29.92% of the area is highly suitable for wheat, teff, and maize, while 70.08% is moderately suitable for wheat but only marginally suitable for teff and maize (Table 9). The majority of marginally and moderately suitable soil textures content ranges for teff and maize crops were located in the south and center of Mansa Watershed, but highly suitable lands were exited in the west and north of the watershed (Fig 6). The findings indicate that the soil textural classes in the study area are favorable for wheat crop production, but there are limitations for teff and maize crops. These findings are consistent with the prior studies conducted by Hussien et al. [57] and Mosisa et al. [55], which also suggested that the soil textures in the study area are moderately appropriate for crop production. Moreover, Motuma et al. [54] found that the soil texture has good potential for the production of wheat and sorghum crops, further supporting the suitability of the soil textures for wheat cultivation in the Mansa watershed.

The elevation map of the watershed was reclassified into four classes to assess its suitability for the selected major cereal crops. The reclassified elevation map reveals that 52.59% of the watershed is highly suitable, 37.21% is moderately suitable, 6.43% is marginally suitable, and 3.76% is not suitable for the selected crops (Table 10). The majority of highly suitable elevations of the lands for teff and maize crops was located in the west, center, and east of the watershed, but moderately suitable lands were exited in the north and east and not suitable lands were in the south of the watershed (Fig 7).The analysis indicates that a significant portion of the watershed has the potential for the production of all selected cereal crops. This finding aligns with the study conducted by Moissa et al. [55], which found that the elevation of the Dedissa watershed was highly suitable for maize cultivation. However, other studies by [49] and Debisa et al. [58] suggested that elevation plays a crucial role in determining cereal crop production. Overall, the reclassified elevation map demonstrates that the majority of the Mansa watershed has favorable conditions for the cultivation of the selected cereal crops

The slope of the Mansa watershed ranges from 0.07 to 35.85%. To assess its suitability for the selected cereal crops, the slope values were reclassified, resulting in a slope suitability map for the study area. The reclassified slope map indicates that 19.45% of the area is highly suitable, 48.40% is moderately suitable, 27.89% is marginally suitable, and 4.26% is not suitable for teff and maize crops. For wheat, 57.51% of the area is highly suitable, 38.22% is moderately suitable, and 4.26% is marginally suitable (Table 11). The majority of highly suitable slope ranges for teff and maize crops was located in the north of Mansa Watershed, but not suitable lands were found in the south of the watershed (Fig 8). The analysis reveals that slope is a limiting factor for teff and maize production in the study area, with only a small portion being highly suitable. However, the majority of the watershed is highly suitable for wheat crop production. This finding

Table 9. Spatial variation of soil texture.

| Criteria | Class | Suitability level | Value | Area (km²) | Area (%) | Crop type |
|---|---|---|---|---|---|---|
| Soil texture | Clay | Highly suitable | 1 | 306.45 | 29.92 | Wheat |
| | Clay loam | Moderately suitable | 2 | 717.79 | 70.08 | |
| | Clay | Highly suitable | 1 | 306.45 | 29.92 | Teff and maize |
| | Clay loam | Marginally suitable | 3 | 717.79 | 70.08 | |

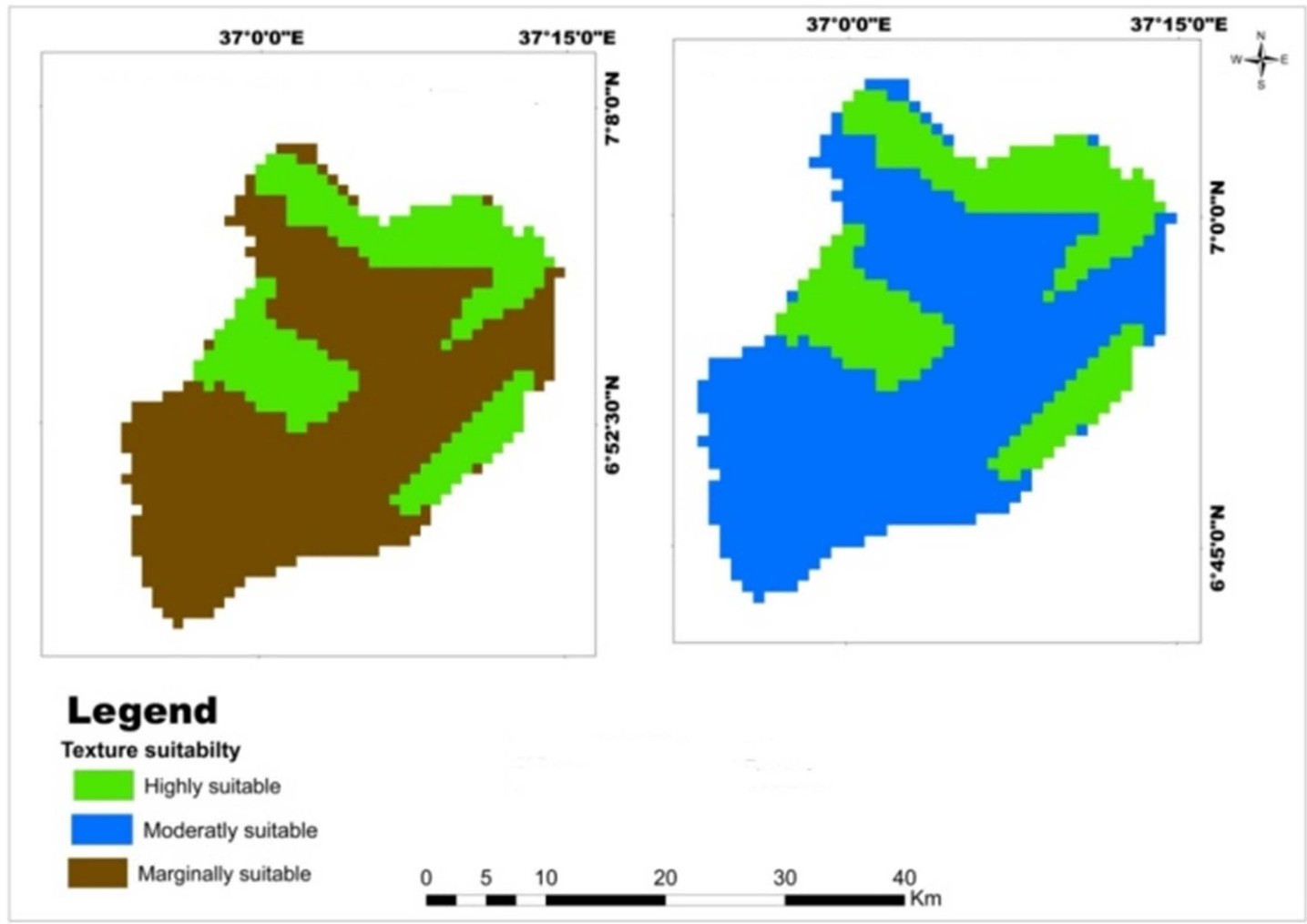

**Fig 6. Land suitability map of classified soil texture for teff, maize, and wheat.**

**Table 10. Spatial variation of elevation.**

| Criteria | Class | Suitability level | Value | Area (km²) | Area (%) | Crop type |
|---|---|---|---|---|---|---|
| Elevation | 1600–2200 | High suitable | 1 | 538.65 | 52.59 | Teff, wheat and maize |
| | 1000–1600/ 2200–2400 | Moderately suitable | 2 | 381.12 | 37.21 | Teff, wheat and maize |
| | 2400–2763 | Marginally suitable | 3 | 65.86 | 6.43 | Teff, wheat and maize |
| | 607–1000 | Not suitable | 4 | 38.51 | 3.76 | Teff, wheat and maize |

is consistent with the study conducted by Kahsay et al. [59], which identified slope as a major constraint for rain-fed teff crop production in the degraded semi-arid highlands of Northern Ethiopia. Mulugeta [60] also found that wheat and maize yields are restricted to steep slope locations with shallow soil depth coverage, which are unfavorable for both crops in Ethiopia's Legambo woreda [61]. Similarly, Rabia [62] identified slope as a limiting factor for teff cultivation in the Kilte

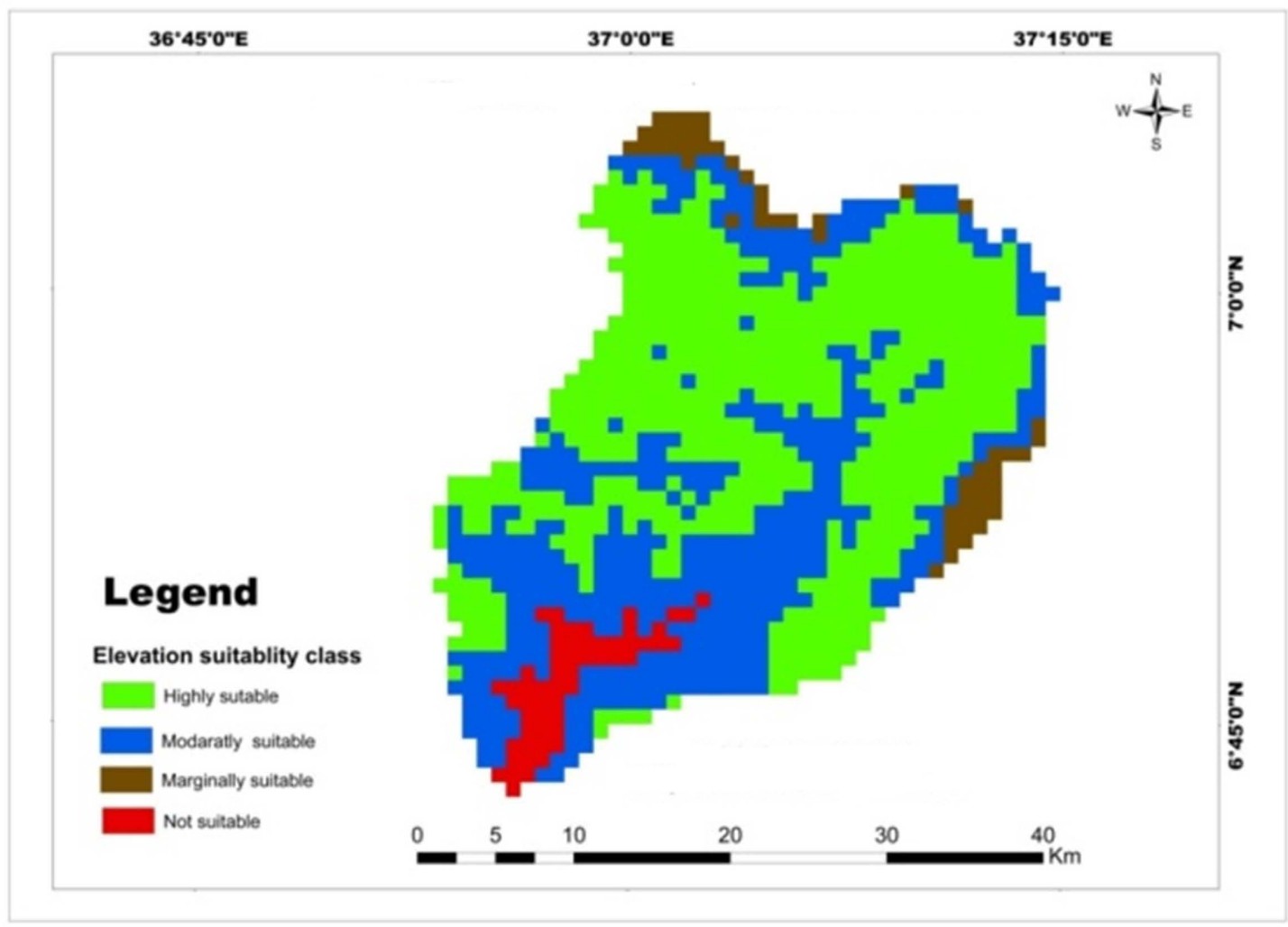

**Fig 7. Land suitability map of classified elevation for teff, maize, and wheat.**

**Table 11. Spatial variation of slope.**

| Criteria | Class (%) | Suitability level | Value | Area (km²) | Area (%) | Crop type |
|---|---|---|---|---|---|---|
| Slope | 0–7 | High suitable | 1 | 199.22 | 19.45 | Teff and maize |
| | 7–15 | Moderately suitable | 2 | 495.73 | 48.40 | |
| | 15–25 | Marginally suitable | 3 | 285.66 | 27.89 | |
| | >25 | Not suitable | 4 | 43.63 | 4.26 | |
| | 0–13 | High suitable | 1 | 589.04 | 57.51 | Wheat |
| | 13–25 | Moderately suitable | 2 | 391.46 | 38.22 | |
| | 25–35.7 | Marginally suitable | 3 | 43.63 | 4.26 | |

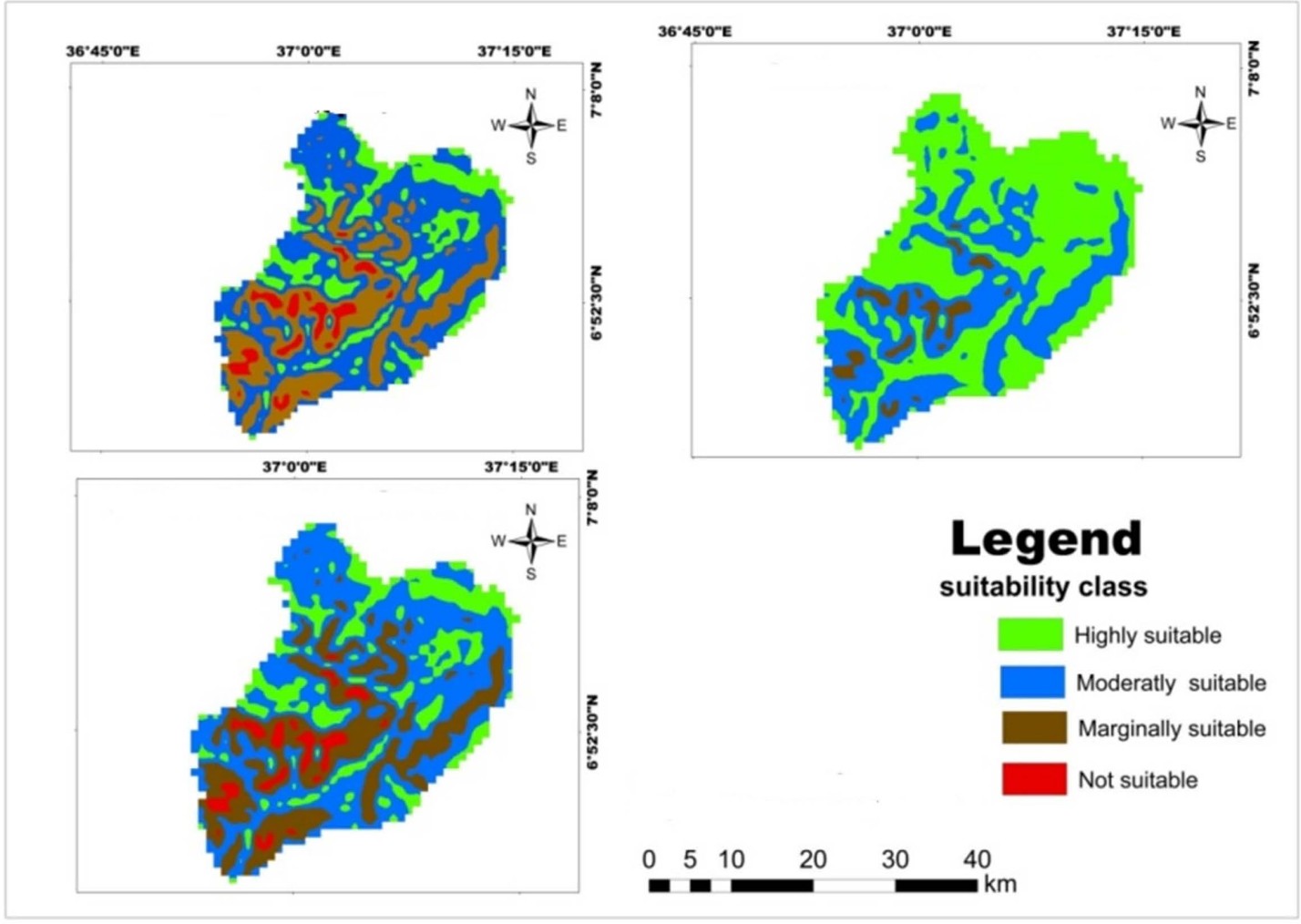

**Fig 8. Land suitability map of classified slope for teff, maize, and wheat crops.**

Awulaelo district. This result aligns with a study by Esa and Assen [36], which highlighted the significant influence of slope on teff production in the Northwest highlands of Ethiopia. Hence, the analysis demonstrates that slope plays a crucial role in determining the suitability of the Mansa watershed for the selected cereal crops, with wheat being more compatible with the existing slope conditions compared to teff and maize. In our study, the production of wheat was highly suitable at lower slope class of the watershed than upper slopes which was comparable to finding by Alemayehu [51] in Sinana Research Site of Southeastern Ethiopia.

The precipitation value is reclassified into suitability class for selected cereal crops. The rainfall variation of the study area showed that 79.6% of the area is marginally suitable for maize, but 100% of the area is not suitable for teff and wheat crops (Table 12). The majority of marginally suitable rainfall ranges for teff and maize crops was located in the east and center of Mansa Watershed, but not suitable lands were found in all parts of the watershed (Fig 9). Similarly Mosisa et al. [56] were reported that rainfall of the Dedissa watershed was unsuitable for maize production. The study conducted by

**Table 12. Spatial variation of rainfall.**

| Criteria | Class (mm) | Suitability level | Value | Area (km²) | Area (%) | Crop type |
|---|---|---|---|---|---|---|
| Rainfall | 1529–1600 | Marginally Suitable | 3 | 815.599 | 79.6386 | Maize |
| | >1600 | Not suitable | 4 | 208.526 | 20.3614 | |
| Rainfall | 1529–1663 | Not suitable | 4 | 1024.23 | 100.0 | Teff and Wheat |

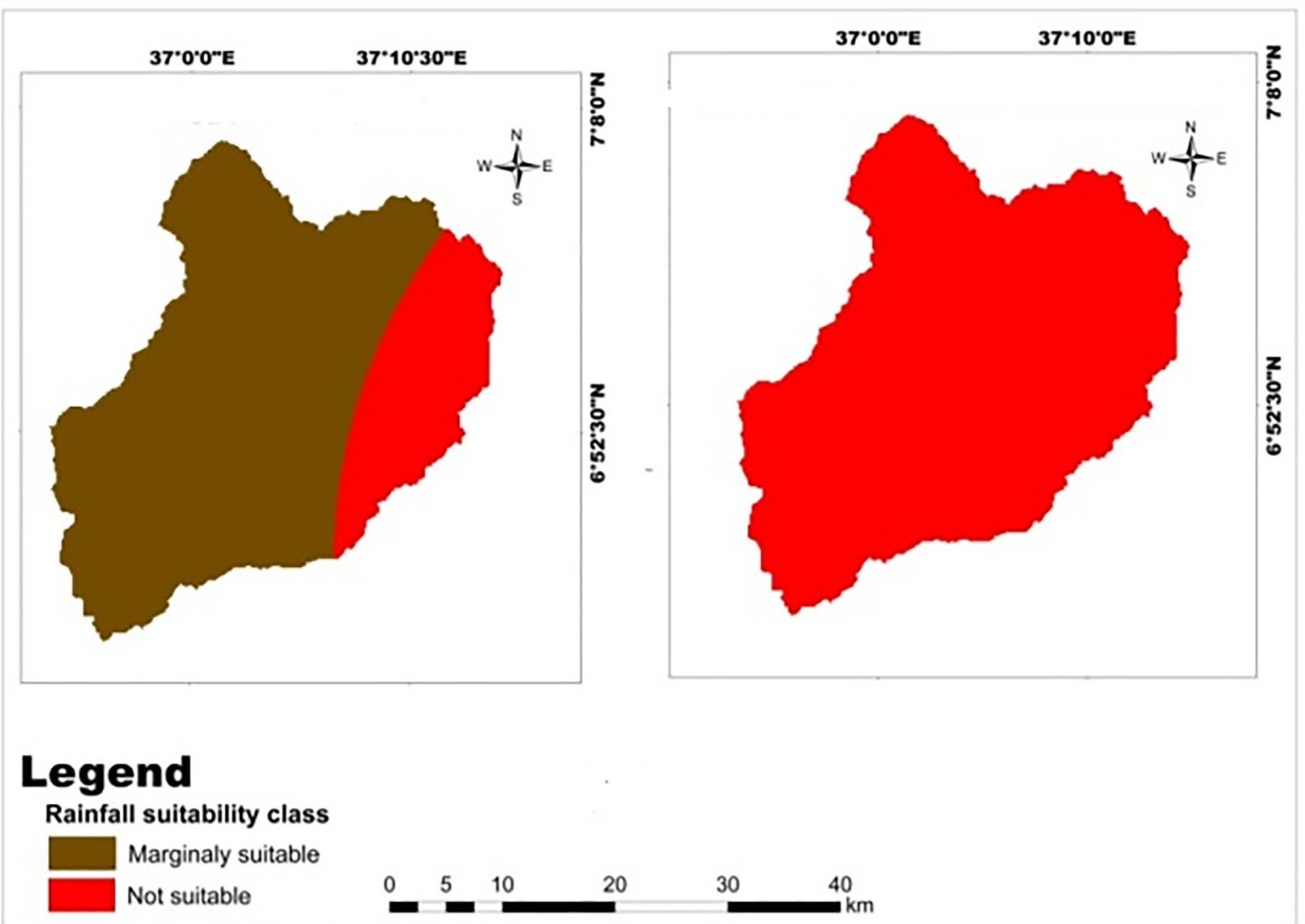

**Fig 9. Land suitability map of classified rainfall for teff, maize, and wheat crops.**

Girmay et al. [53] in Gateno watershed showed that rainfall and temperature were the main limiting factor for the production of wheat and barley crops.

The Mansa watershed experiences an annual average temperature ranging from 16.4°C to 20.5°C. The reclassified temperature map indicates that 5.38% of the watershed is highly suitable, 16.87% is moderately suitable, and 77.77% is marginally suitable for wheat crops (Table 13).The majority of marginally suitable temperature ranges for teff and maize crops was located in center and eastern parts of Mansa Watershed, but a small parcel of highly suitable lands were found in east most

**Table 13. Spatial variation of temperature.**

| Criteria | Class (°C) | Suitability level | Value | Area (km²) | Area (%) | Crop type |
|---|---|---|---|---|---|---|
| Temperature | 16.4–18.4 | Highly suitable | 1 | 54.95 | 5.38 | Wheat |
| | 18.4–19.4 | Moderately suitable | 2 | 172.75 | 16.87 | |
| | 19.4–20.5 | Marginally suitable | 3 | 796.39 | 77.77 | |
| | 16–20.5 | Highly suitable | 1 | 1024.23 | 100.0 | Teff and maize |

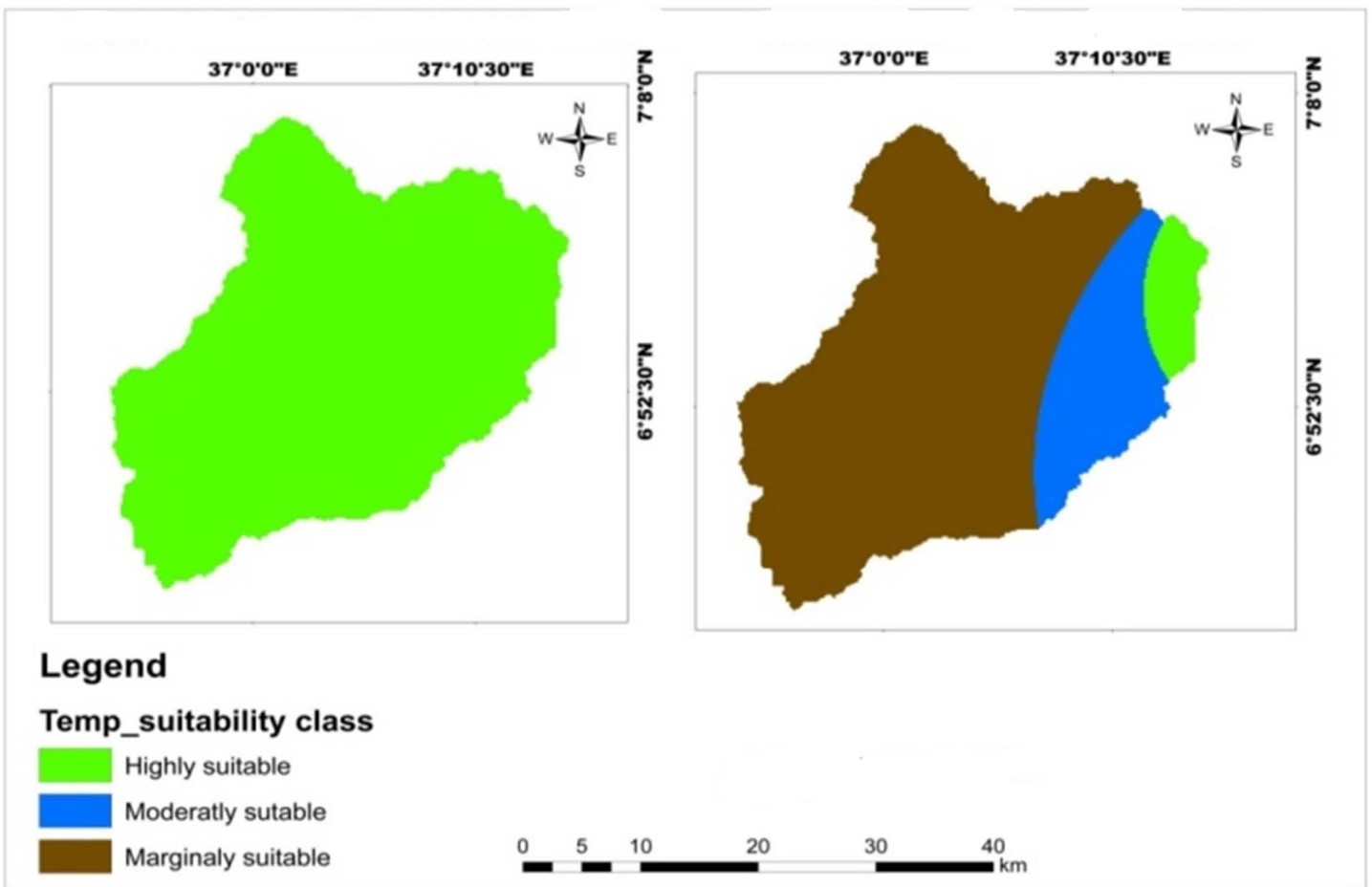

**Fig 10. Land suitability map of classified temperature for teff, maize, and wheat crops.**

of the watershed (Fig 10). This suggests that temperature is the main limiting factor for wheat crop production in the study area. The optimal temperature range required for wheat crops is 14.9–18.4°C, which falls within the highly suitable category. However, the majority of the study area is classified as marginally suitable for wheat cultivation. This finding is consistent with a study by Hailu et al. [63], which identified temperature as the main limiting factor for wheat and barley crops in the Kabe sub watershed. On the other hand, the Mansa watershed is highly suitable for teff and maize crops, as indicated by the reclassified temperature map (Table 14; Fig 13). The analysis suggests that the temperature conditions in the study area are favorable for teff and maize production. However, a study by Yohannes and Soromessa [42] reported that temperature

**Table 14. LULC types of the study area.**

| LULC class | Area (km²) | % |
|---|---|---|
| Forest land | 395.11 | 38.5772 |
| Shrubs cover areas | 24.38 | 2.380011 |
| Grassland | 123.23 | 12.03149 |
| Cropland | 480.64 | 46.92819 |
| Vegetation cover | 0.06 | 0.005595 |
| Bare land | 0.11 | 0.011123 |
| Settlement area | 0.68 | 0.06631 |
| Open water | 0.001 | 0.000287 |
| Total | 1024.23 | 100.0 |

was the most limiting factor for cereal crop production, which contrasts with the findings of this study. Therefore, the analysis reveals that temperature plays a significant role in determining the suitability of the Mansa watershed for wheat, teff, and maize crops. While the temperature conditions in the study area are not optimal for wheat cultivation, they are suitable for teff and maize production.

**Land use/land cover.** For each land use and land cover class, 30 signatures were collected, and the study analysis classified the Sentinel-2A satellite image into eight land cover types: cropland, forest land, grassland, vegetation covers, shrub covers, bare land, settlement area, and open water. The maximum likelihood classification of the image revealed that cropland occupies the largest portion of the area in the northern parts of the watershed (Fig 11) which were highly suitable for production for the selected crops for these analyses (Fig 12), while water bodies cover a relatively small area compared to the other land cover types (Table 14).

**Accuracy assessment of LULC classification.** The accuracy assessment of the classification resulted in an overall accuracy of 88.7% and a kappa coefficient of 0.87 (Table 15). These values indicate a high level of accuracy and reliability in the classification results, providing a solid foundation for further analysis. The satisfactory classification accuracy assessment results align with previous studies [46,47], further validating the findings. The land use and land cover (LULC) types play a crucial role in determining the adaptability of various important cereal crops. Therefore, the LULC of the watershed was categorized into four groups based on crop suitability, taking into account the impact of LULC on the growth and productivity of these crops (Table 16; Fig 12).

## Physical land suitability analysis results for selected major cereal crops

Results of a physical land suitability analysis for teff, maize, and wheat crops were obtained through a weighted overlay analysis of ten factors. The analysis revealed that in the study area, 61.18% (626.5 km²) and 38.82% (397.52 km²) of the land were moderately suitable and marginally suitable for teff crops respectively (Table 17; Fig 13). The majority of highly suitable LULC type for teff and maize crops was located in the north and east of Mansa Watershed, but not suitable lands were found in the south and southwest of the watershed Figs 13,14). However, no areas were classified as highly suitable (S1) or not suitable (N1) for teff cultivation. This indicates that the study area has significant limitations in terms of rainfall, soil organic matter, pH, soil texture, elevation, soil drainage, and soil depth. These findings align with previous research conducted in Northern Ethiopia's degraded semi-arid highlands by Kahsay et al. [59] and the Gelda catchment by Esa and Assen [36]. However, a different study in Kilte Awulaelo Woreda, Tigray region showed that most of the study area was permanently unsuitable for teff and maize production Hishe and Assen [64]. For wheat production, 50.063% (512.703 km²), 13.9% (142.398 km²), and 36.03% (368.99 km²) of the land in the study area were assessed as moderately suitable, marginally suitable, and not suitable, respectively (Table 17; Fig 15). Similar findings

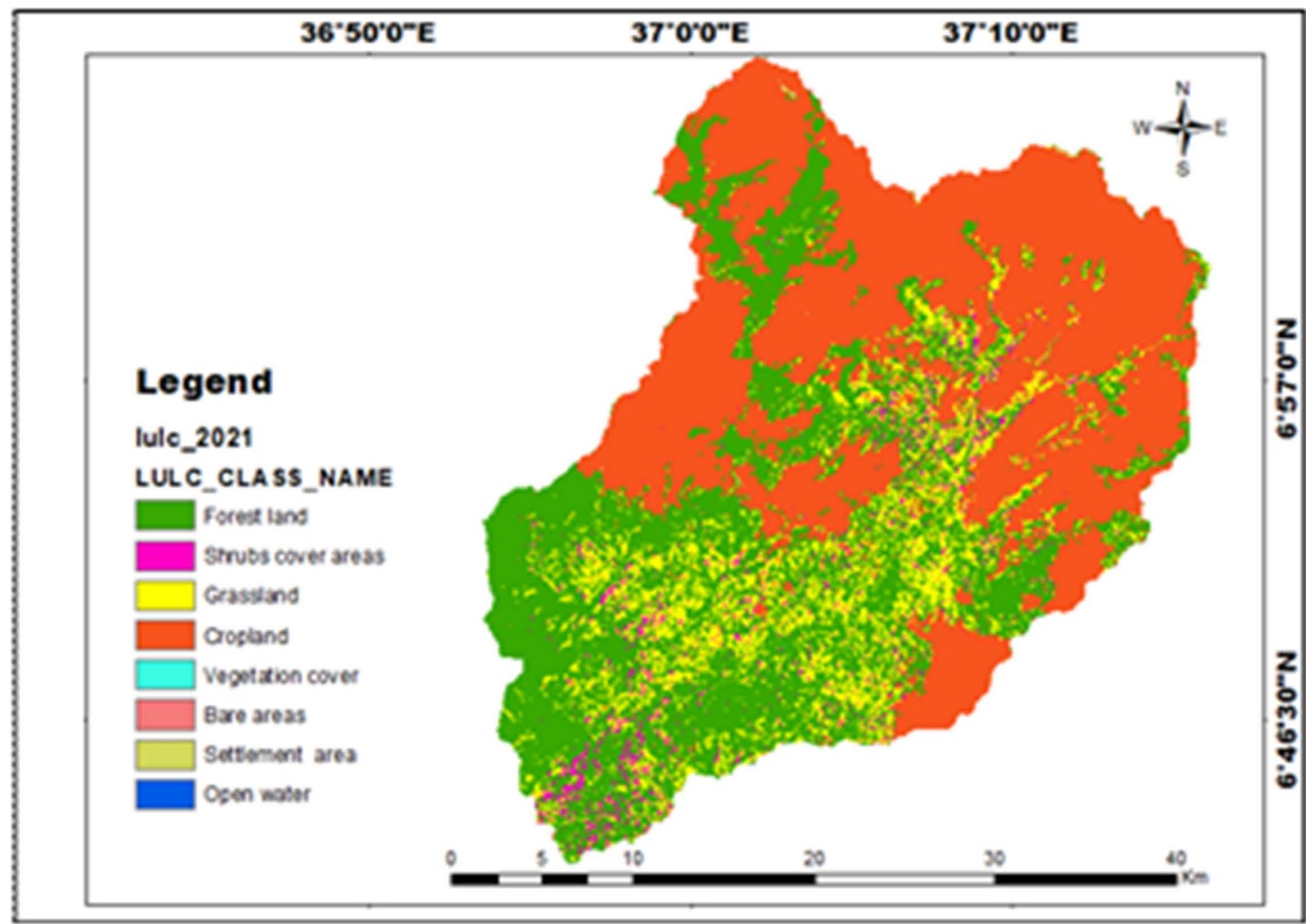

**Fig 11. LULC types and their area proportions for Mansa Watershed.**

were reported in South Wello, Yikalo sub-watershed, Kabe sub-watershed, Andit Tid watershed, Anjeni watershed [65], and Awulaelo District in Tigray region (Motuma et al. [54], Fekadu and Negese [49], Hailu et al. [63], Yohannes and Soromessa [42], Rabia [62]). These studies identified varying proportions of moderately suitable, marginally suitable, and highly suitable areas for wheat production. Regarding maize cultivation, 29.65% (303.62 km$^2$), 52.81% (540.78 km$^2$), and 17.55% (179.72 km$^2$) of the land in the study area were classified as moderately suitable, marginally suitable, and not suitable, respectively (Table 17; Fig 14). Similar findings were reported in the Dedissa watershed, Bilate Alaba Sub-watershed, Abobo region of western Ethiopia, and Dabo Hana district, indicating varying proportions of suitability for maize crop production [4,53,56,58]. Overall, the results of the analysis demonstrate the suitability of the study area for teff, wheat, and maize cultivation, but also highlight the need for addressing the limitations identified through sustainable land management strategies.

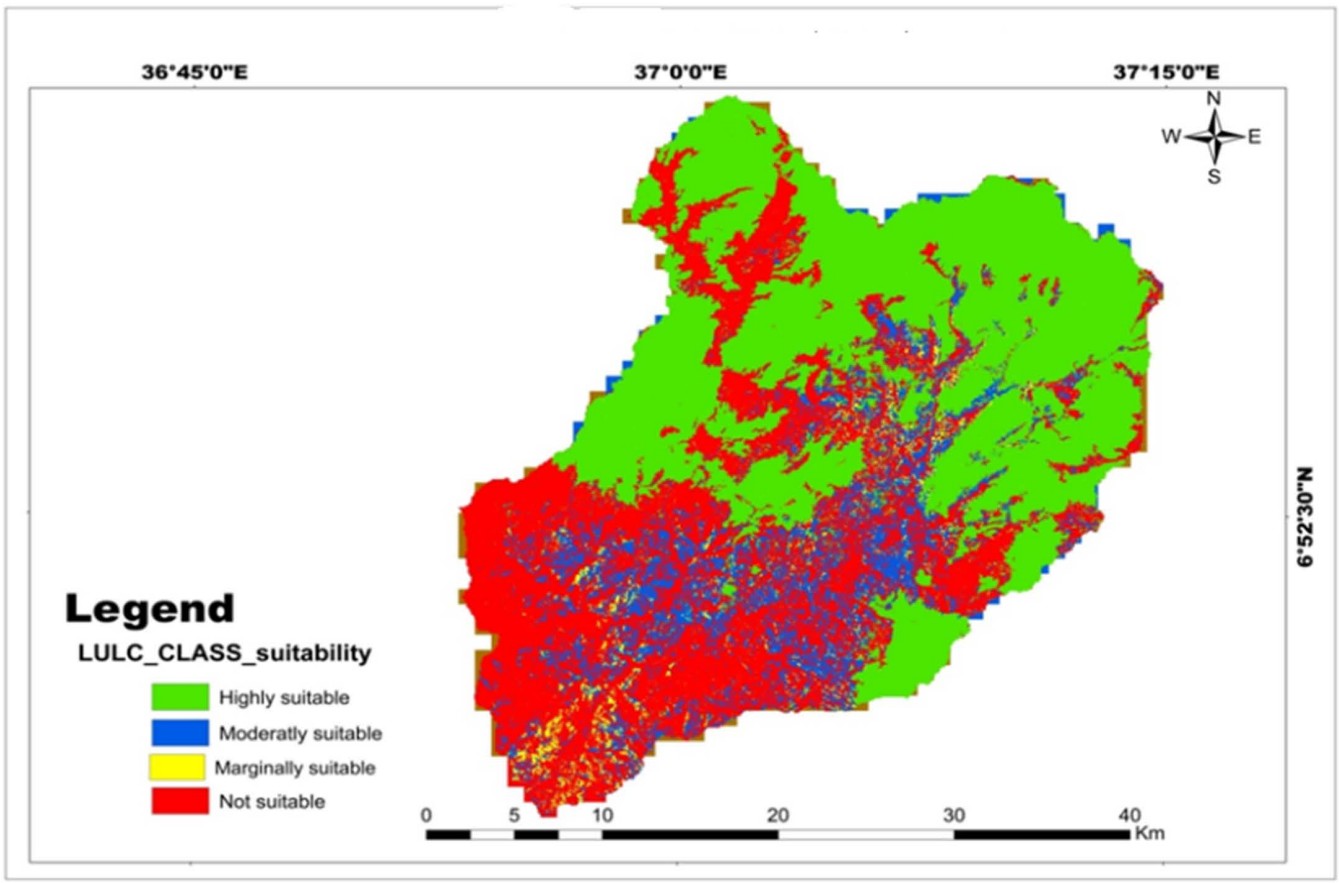

**Fig 12. Land suitability map of classified LULC for teff, maize, and wheat crops.**

**Table 15. Accuracy assessment.**

| Reference Data | | | | | | | | | | |
|---|---|---|---|---|---|---|---|---|---|---|
| Classified data | Forest land | Shrubs cover areas | Grassland | Cropland | Vegetation cover | Bare land | Settlement areas | Open water | Total | User accuracy |
| Forest land | 27 | 1 | 1 | 0 | 2 | 0 | 0 | 0 | 31 | 87.1 |
| Shrubs cover areas | 2 | 28 | 1 | 0 | 0 | 0 | 0 | 0 | 31 | 90.3 |
| Grassland | 0 | 1 | 27 | 2 | 0 | 0 | 0 | 0 | 30 | 90.0 |
| Cropland | 0 | 0 | 1 | 26 | 0 | 6 | 1 | 0 | 34 | 76.5 |
| Vegetation cover | 1 | 0 | 0 | 0 | 28 | 0 | 0 | 0 | 29 | 96.6 |
| Bare land | 0 | 0 | 0 | 2 | 0 | 22 | 4 | 0 | 28 | 78.6 |
| Settlement area | 0 | 0 | 0 | 0 | 0 | 2 | 25 | 0 | 27 | 92.6 |
| Open water | 0 | 0 | 0 | 0 | 0 | 0 | 0 | 30 | 30 | 100.0 |
| Total | 30 | 30 | 30 | 30 | 30 | 30 | 30 | 30 | 240 | |
| Producer accuracy | 90.0 | 93.3 | 90.0 | 86.7 | 93.3 | 73.3 | 83.3 | 100.0 | | |

**Table 16. LULC class with respective suitability for the selected crop.**

| LULC class | Suitability level | Value | Area (km²) | Percent |
|---|---|---|---|---|
| Cropland | High suitable | 1 | 480.64 | 46.93 |
| Grassland | Moderately suitable | 2 | 123.22 | 12.03 |
| Shrubs cover areas/bare areas | Marginally suitable | 3 | 24.49 | 2.39 |
| Forest land/vegetation/settlement areas/open water | Not suitable | 4 | 395.85 | 38.65 |
| | Total | | 1024.23 | 100.0 |

**Table 17. Area coverage of land suitability for selected cereal crops in the watershed.**

| Suitability Classes | Maize | | Teff | | Wheat | |
|---|---|---|---|---|---|---|
| | Area(km²) | (%) | Area(km²) | (%) | Area(km²) | (%) |
| S2 | 303.62 | 29.65 | 626.50 | 61.18 | 512.703 | 50.063 |
| S3 | 540.78 | 52.81 | 397.52 | 38.82 | 142.398 | 13.9 |
| N1 | 179.72 | 17.55 | – | – | 368.99 | 36.03 |

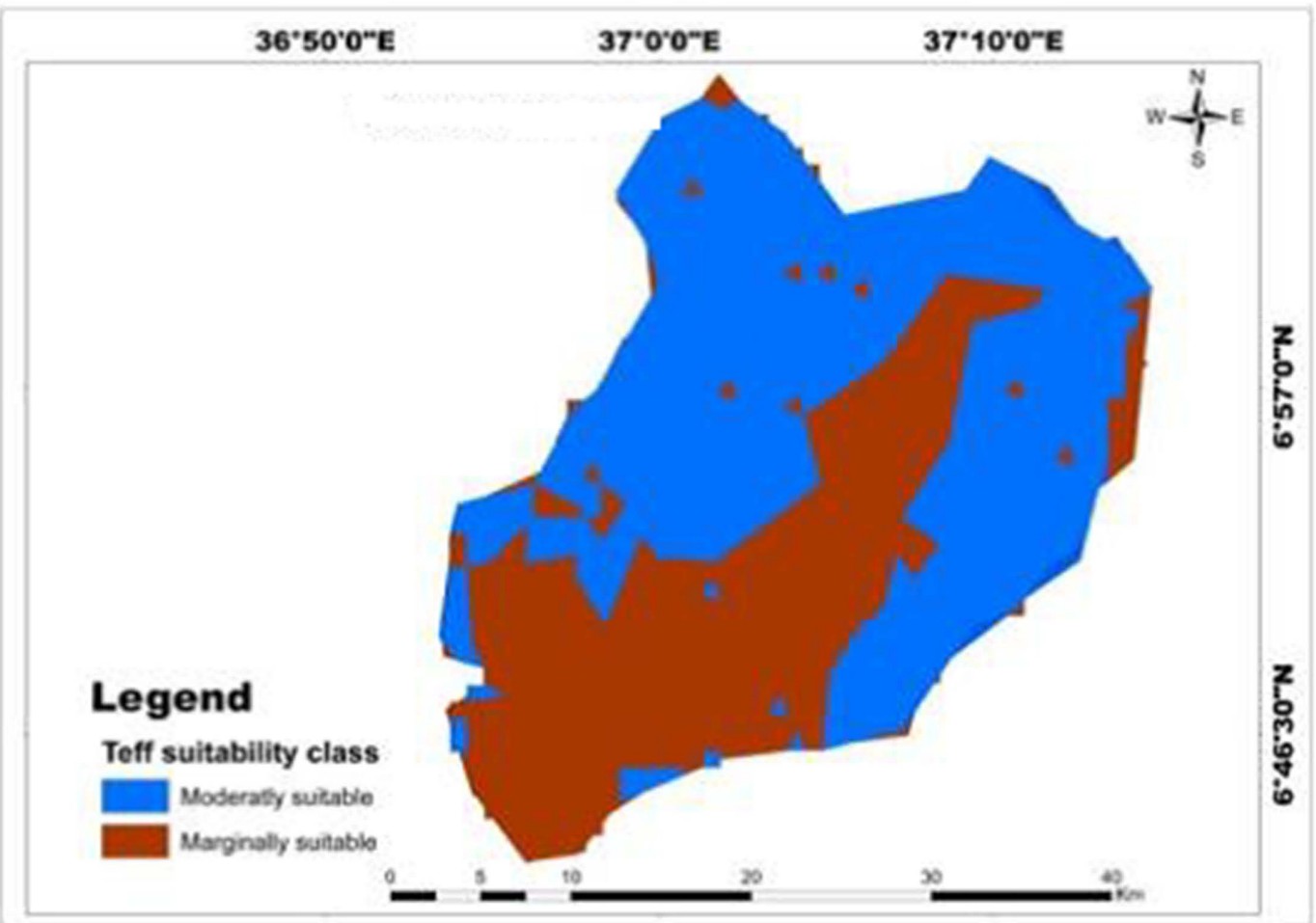

**Fig 13. Land suitability map classified soil depth for teff.**

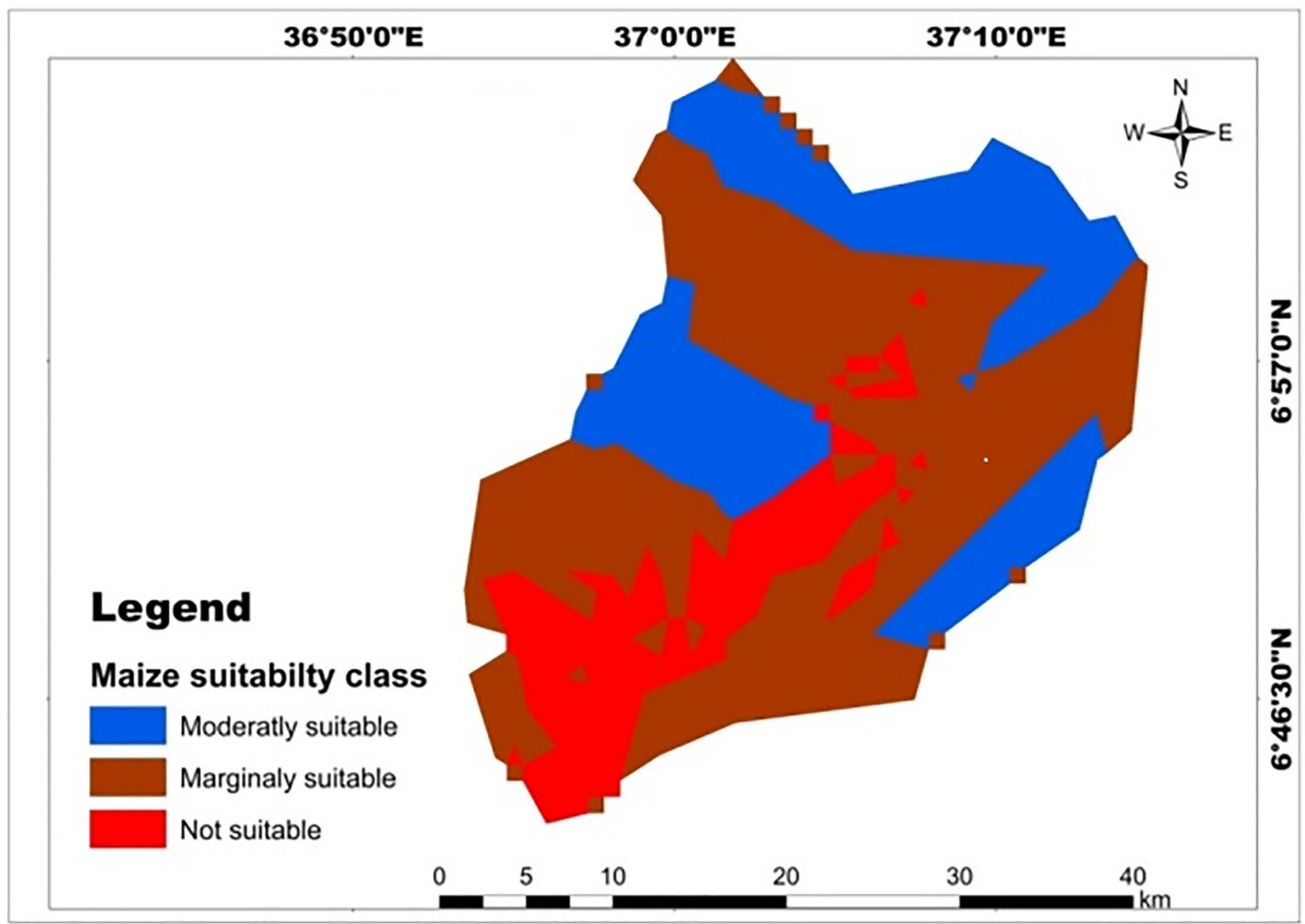

**Fig 14. Land suitability map classified soil depth for maize crop.**

**Land allocation for major cereal crops using current suitability maps**

The suitable land allocation for selected land use types (LUTs) in the watershed, based on their best suitability classes is presented in Table 18. Approximately 44.2% (442.1 km²) and 10.1% (110.2 km²) of the study area are classified as moderately suitable (S2) and marginally suitable (S3) for all specified land use categories, respectively (Fig 16). This finding aligns with similar studies conducted in other regions [25,65], indicating that different LUTs can compete for the same land with equal suitability ratings. The remaining area exhibits mixed suitability for the analyzed crops. For instance, about 2.7% (27.9 km²) of land is moderately suitable (S2) for teff and wheat production but only marginally suitable (S3) for maize crops. Similarly, approximately 0.51% (1.53 km²) of land is marginally suitable (S3) for maize and teff production but moderately suitable (S2) for wheat cultivation. This indicates that farmers have the flexibility to choose land use types that best suit their requirements [66]. However, these findings also highlight

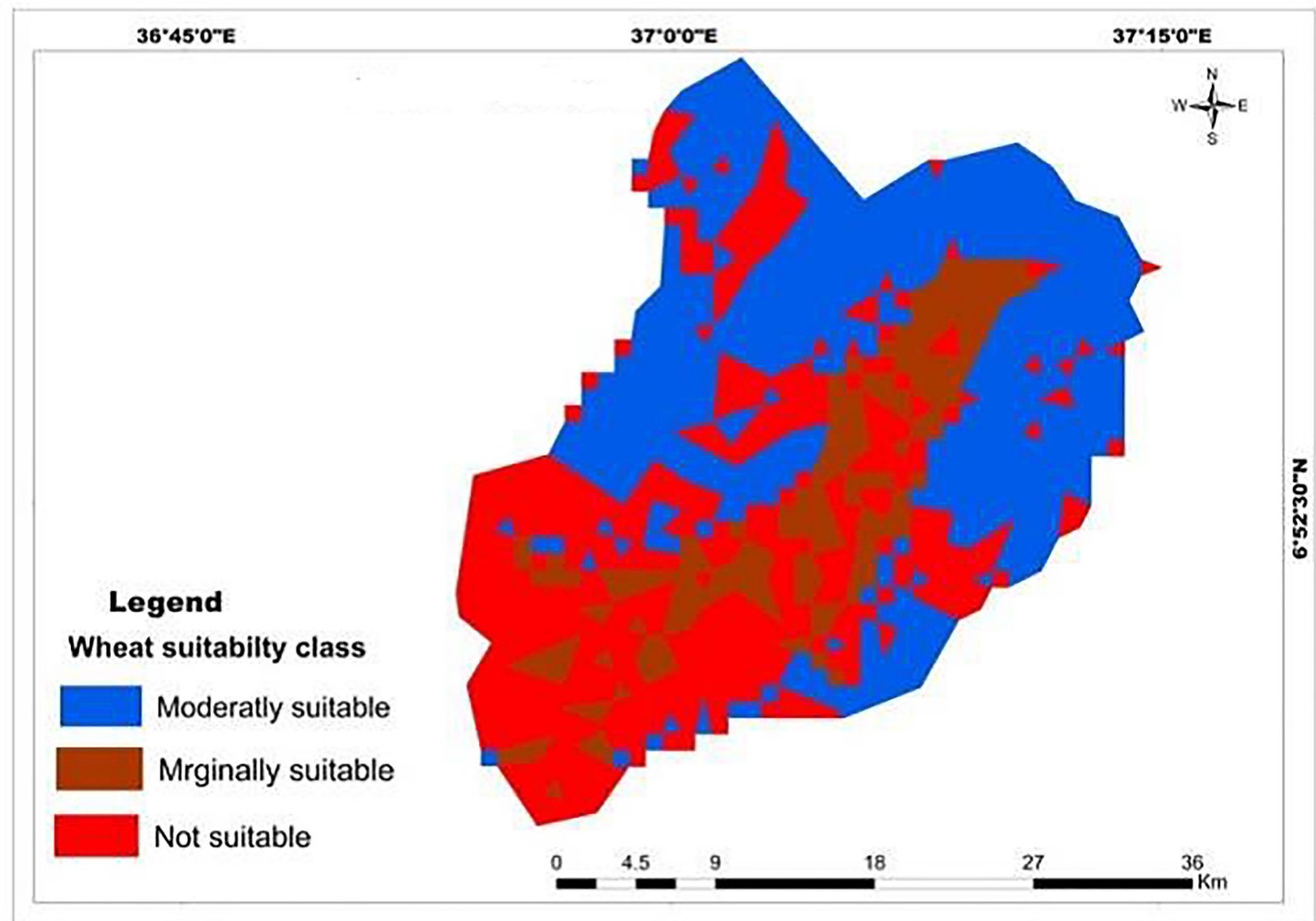

**Fig 15. Land suitability map classified soil depth for wheat crop.**

the competition for land among various land use alternatives with different suitability levels [58,60]. The remaining suitable land allocation classes represent small patches of land across the study area (Table 18). In the future work, the production of these crops could integrate socioeconomic factors like market and irrigation facilities in the study area which are vital for agricultural planning [67].

## Conclusions

This study utilized GIS and AHP approaches to assess the suitability of land for selected cereal crops in the Mansa watershed, aiming for sustainable agriculture. The results revealed that there were no highly suitable lands for the production of the selected major cereal crops. Instead, most of the lands in the watershed were classified as moderately and marginally suitable for teff, wheat, and maize crops. The study also identified significant differences in land suitability among different land units for each cereal crop. Teff cultivation was found to be relatively better suited compared to the current land use practices of maize and wheat. However, wheat and maize cultivation still remain viable options.

**Table 18. Suitable land allocation and area coverage for selected cereal crops.**

**Overall Suitability Analysis**

| SN | Code | Area (km²) | Area (%) |
|---|---|---|---|
| 1 | S2_Mz | 7.87 | 0.7 |
| 2 | S2_TF | 0.054 | 0.005 |
| 3 | S2_Wt | 0.39 | 0.03 |
| 4 | S2_TF,Wt | 2 | 0.2 |
| 5 | S2_Mz.TF | 0.28 | 0.02 |
| 6 | S2_Mz,Wt | 1.1 | 0.1 |
| 7 | S2_Mz,TF,Wt | 442.1 | 44.2 |
| 8 | S2_Mz,Wt,N1_TF | 1.1 | 0.1 |
| 9 | S2_TF,N1_Mz,Wt | 18.6 | 1.8 |
| 10 | S2_ TF, S3_Mz,Wt | 2.1 | 0.2 |
| 11 | S2_Wt,TF, N1_Mz | 3.8 | 0.3 |
| 12 | S2_TF,Wt, S3_Mz | 27.9 | 2.7 |
| 13 | S2_Mz, N1_Wt,TF | 1.35 | 0.13 |
| 14 | S2_Wt, S3_Mz | 0.13 | 0.1 |
| 15 | S2_Wt,N1_Mz,TF | 0.7 | 0.07 |
| 16 | S2_Wt, S3_Mz, N1_TF | 7.1 | 0.7 |
| 17 | S2_Wt,S3_TF,Mz | 1.53 | 0.15 |
| 18 | S3_Mz,Wt | 31.9 | 3.1 |
| 19 | S3_Mz | 0.2 | 0.02 |
| 20 | S3_Mz, N1_Wt,TF | 1.3 | 0.13 |
| 21 | S3_Mz,TF,Wt | 110.2 | 10.1 |
| 22 | S3_Mz,Wt,N1_TF | 10.2 | 1.02 |
| 23 | S3_TF | 23.3 | 2.3 |
| 24 | S3_Wt,TF,N1_Mz | 0.19 | 0.01 |
| 25 | S3_Wt,N1_TF,Mz | 0.12 | 0.01 |
| 26 | N1_Mz,TF,Wt | 6.2 | 0.6 |
| 27 | Restricted | 322 | 32. |
|  | Total | 1024.23 | 100.0 |

The study identified soil pH, organic matter, and rainfall as the main limiting factors for the production of teff, wheat, and maize crops. Proper management measures are necessary to address these limitations and maintain the productivity of the farmlands.

It was found that there were no highly suitable lands for the production of selected major cereal crops. Instead, most of the lands in the watershed were classified as moderately and marginally suitable. In the future, sustainable land management interventions will be essential for enhancing land suitability and improving soil properties. Additionally, the analysis should encompass irrigation facilities, market access, processing industries, and other socioeconomic factors to offer a wider range of options for stakeholders and policy decision makers. Lastly, appropriate conservation measures should be implemented to maximize crop production and sustain the soil's productive capacity in areas classified as moderately and marginally suitable for the selected cereal crops.

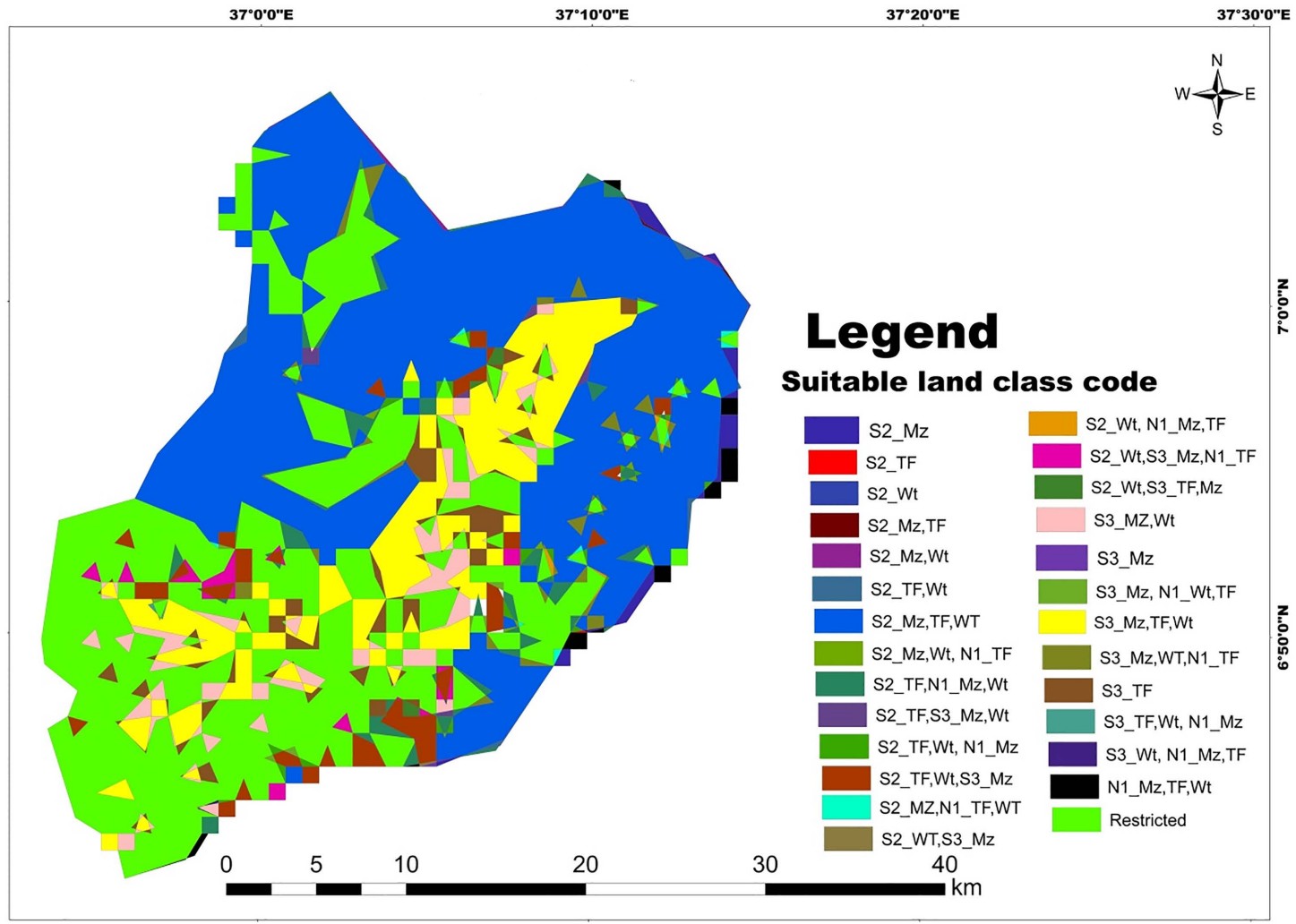

**Fig 16.  Appropriate Land Allocation Map with their Respective Degree of Suitability in Mansa watershed (2022) (Note: S2 = moderately suitable, S3 = marginally suitable, N1 = currently not suitable, Tf = Teff, Mz = Maize, Wt = Wheat).**

## Acknowledgments

The authors express their thanks to Agricultural and Natural Resource office of Dawuro zone as well as other natural resource experts is also highly acknowledged.

## Author contributions

**Conceptualization:** Wakshum Shiferaw, Wude Taye, Genaye Tsegaye.

**Data curation:** Wude Taye.

**Formal analysis:** Wude Taye.

**Funding acquisition:** Wude Taye.

**Investigation:** Wakshum Shiferaw, Wude Taye, Genaye Tsegaye.

**Methodology:** Wakshum Shiferaw, Wude Taye, Genaye Tsegaye.

**Software:** Wude Taye.

**Supervision:** Wakshum Shiferaw, Genaye Tsegaye.

**Validation:** Wakshum Shiferaw, Wude Taye, Genaye Tsegaye.

**Visualization:** Genaye Tsegaye.

**Writing – original draft:** Wude Taye.

**Writing – review & editing:** Wude Taye, Genaye Tsegaye.

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
