## [Decision Letter · Decision Letter 0]

27 Dec 2023

PONE-D-23-35036“Unlocking Agricultural Potential: Analyzing Land Suitability for Key Cereal Crops in Mansa Watershed, Southwest Ethiopia”PLOS ONE

Dear Dr. Shiferaw,

Thank you for submitting your manuscript to PLOS ONE. After careful consideration, we feel that it has merit but does not fully meet PLOS ONE’s publication criteria as it currently stands. Therefore, we invite you to submit a revised version of the manuscript that addresses the points raised during the review process.

We look forward to receiving your revised manuscript.

Kind regards,

Tunira Bhadauria, Ph.D.

Academic Editor

PLOS ONE

Journal Requirements:

Reviewers' comments:

Reviewer's Responses to Questions

**Comments to the Author**

1. Is the manuscript technically sound, and do the data support the conclusions?

Reviewer #1: Yes

Reviewer #2: Yes

2. Has the statistical analysis been performed appropriately and rigorously? 

Reviewer #1: Yes

Reviewer #2: I Don't Know

3. Have the authors made all data underlying the findings in their manuscript fully available?

Reviewer #1: Yes

Reviewer #2: Yes

4. Is the manuscript presented in an intelligible fashion and written in standard English?

Reviewer #1: Yes

Reviewer #2: Yes

5. Review Comments to the Author

Reviewer #1: This study utilized GIS and AHP approaches to assess the suitability of land for selected

cereal crops in the Mansa watershed, aiming for sustainable agriculture are novel study

The results revealed that there were no highly suitable lands for the production of the selected major cereal crops. Instead, most of the lands in the watershed were classified as moderately and marginally

suitable for teff, wheat, and maize crops. Result are acceptable

A abstract or conclusive figure must be added in abstract

For methodology a figure musts be added which provide step by step information about experiment

Material method must be rewritten

Future prospective must be added

Reviewer #2: Though it was a nice study to go through, I have few questions and suggestions.

1. What is the novelty of the study as similar land suitability analysis were undertaken/repeated in other locations?

2. Do such studies match with the ground reality?

3. Add and compare the results of the manuscript attached.

6. PLOS authors have the option to publish the peer review history of their article (what does this mean? ). If published, this will include your full peer review and any attached files.

**Do you want your identity to be public for this peer review?** For information about this choice, including consent withdrawal, please see our Privacy Policy .

Reviewer #1: **Yes: ** Kuldip Jayaswall

Reviewer #2: No

---

## [Author Response · Author response to Decision Letter 1]

16 Feb 2024

Response to editor on a manuscript entitled “Unlocking Agricultural Potential: Analyzing Land Suitability for Key Cereal Crops in Mansa Watershed, Southwest Ethiopia” (PONE-D-23-35036).

We would like to thank the editor for his very useful and constructive comments that have helped us to substantially improve the manuscript confirm the figures submission were created with non-copyrighted sources. We believe to have addressed all comments in the reports. With the specific information requested, we already have information in Table 1 and Table 2 of the manuscript. We have the responses are provided below.

Fig 7. Reclassified elevation map - Software: ArcMap 10.3, ERDAS IMAGINE 2015, IDRISI 17.0, and Google Earth-source https://earthexplorer.usgs, Ethiopian Mapping Agency (base map).

Fig 11. LULC cover of the study area- Sentinel-2A image of 2021 from https://earthexplorer.usgs, Ethiopian Mapping Agency (base map), Software used: ArcMap 10.3, ERDAS IMAGINE 2015, IDRISI 17.0, and Google Earth.

Fig 12. Reclassified LULC map - Sentinel-2A image of 2021 from https://earthexplorer.usgs, Ethiopian Mapping Agency (base map), Software: ArcMap 10.3, ERDAS IMAGINE 2015, IDRISI 17.0, and Google Earth.

Fig 13. Teff suitability map - Ethiopian Mapping Agency, Agricultural Transformation Agency (ATA), Ethiopia National Meteorology Agency (NMA), Software used: ArcMap 10.3, ERDAS IMAGINE 2015, IDRISI 17.0, and Google Earth.

Fig 14. Wheat suitability map- Sources are Ethiopian Mapping Agency (Base map), Agricultural Transformation Agency (ATA) to develop soil OM, pH, Depth, Drainage and Texture map of the study area, Ethiopia National Meteorology Agency (NMA) to interpolate temperature and rainfall data. Software used: ArcMap 10.3, ERDAS IMAGINE 2015, IDRISI 17.0, and Google Earth.

Fig 15. Maize suitability map- Ethiopian Mapping Agency (Base map), Agricultural

Transformation Agency (ATA) to develop soil OM, pH, Depth, Drainage and Texture map of the study area, Ethiopia National Meteorology Agency (NMA) to interpolate temperature and rainfall data. Software used: ArcMap 10.3, ERDAS IMAGINE 2015, IDRISI 17.0, and Google Earth

As aforementioned, more informations is available in table 1 and 2 of the manuscript.

---

## [Decision Letter · Decision Letter 1]

20 Mar 2024

PONE-D-23-35036R1“Unlocking Agricultural Potential: Analyzing Land Suitability for Key Cereal Crops in Mansa Watershed, Southwest Ethiopia”PLOS ONE

Dear Dr. Shiferaw,

Thank you for submitting your manuscript to PLOS ONE. After careful consideration, we feel that it has merit but does not fully meet PLOS ONE’s publication criteria as it currently stands. Therefore, we invite you to submit a revised version of the manuscript that addresses the points raised during the review process.

We look forward to receiving your revised manuscript.

Kind regards,

Nguyen-Thanh Son, Ph.D.

Academic Editor

PLOS ONE

Additional Editor Comments:

• The title of the paper: Please remove ‘Unlocking Agricultural Potential:”. It doesn’t make sensed.

• The abstract is unclear and the authors better highlight the importance of this research, methods (processing steps), and major findings of the study.

• Introduction section: AHP and MCDM are not new algorithms. It has been used in many spatial modeling studies. In other words, the specific innovation regarding the methodology should be addressed. What is methodologically research gap the author want to solve in this study, based on the literature review? Explanations should be added.

• Introduction section, aside from AHP and MCE (Multi-Criteria Evaluation), other methods such as GARP (Genetic Algorithm for Rule-set Prediction), CA-based MCE, and machine-learning based approaches have been commonly applied for land suitability assessment. Please provide advantages and disadvantages of the use of your methods compared to such algorithms.

• Study area and data collection should be presented in separate sections.

• The size of the study region is relatively small, while the spatial resolution of geospatial datasets (e.g., rainfall, soil properties) are coarse. Can you extend the study area to a larger region? The soil map used in this study has a scale of 1:250,000. Please consider using the global soil map (250 m resolution), which is offering free of charge about the world.

• What is the rationale for selecting the study area? Describe more details about the study area as well as major crops being analyzed.

• Table 1: change 30m*30m to 30m×30m.

• Ground data collection, please mention how you design sampling methods for collecting ground reference samples, and accuracy of the ground truth.

• ‘Kappa Coefficient’ should be combined with ‘Accuracy Assessment’.

• In this work, the scores or weights for factors were allocated based on literature and expert opinions. However, the incorporation of expert opinions and results from the literature for suitability analysis reveal limitations, including subjectivity and biases that can introduce uncertainties in the model. Thus, I would suggest that participatory approaches involving multiple experts and local communities or stakeholders to improve the representativeness of collective knowledge while mitigating individual biases to ultimately enhance the overall accuracy of land suitability assessments. Please elaborate on your explanations for these concerns in the introduction or methodology section.

• Elaborate classification methods (unsupervised and supervised algorithms) used for mapping LULC types. For example, how did you select LULC classes, training datasets for model training and validation. Detailed descriptions about classification methods must be addressed.

• There should be a more thorough discussion of uncertainties and cautions about the application of the methodology in other regions in the discussion section. The comparison to related studies should be addressed.

• There are still grammatical issues and inappropriate sentence structures; it is recommended that the authors should hire a professional English editor to go through the editing.

• Some minor corrections: The quality of all figures should be improved. Tittles of figures can be noted in the caption. For example, Figure: Soil depth suitability: (a) reclassified soil depth for Teff and maize, and (b) reclassified soil depth for wheat. Math notations in all equations should follow the format of the journal.

Reviewers' comments:

Reviewer's Responses to Questions

**Comments to the Author**

1. If the authors have adequately addressed your comments raised in a previous round of review and you feel that this manuscript is now acceptable for publication, you may indicate that here to bypass the “Comments to the Author” section, enter your conflict of interest statement in the “Confidential to Editor” section, and submit your "Accept" recommendation.

Reviewer #1: All comments have been addressed

Reviewer #2: (No Response)

2. Is the manuscript technically sound, and do the data support the conclusions?

Reviewer #1: Yes

Reviewer #2: Yes

3. Has the statistical analysis been performed appropriately and rigorously? 

Reviewer #1: Yes

Reviewer #2: N/A

4. Have the authors made all data underlying the findings in their manuscript fully available?

Reviewer #1: Yes

Reviewer #2: Yes

5. Is the manuscript presented in an intelligible fashion and written in standard English?

Reviewer #1: Yes

Reviewer #2: Yes

6. Review Comments to the Author

Reviewer #1: Manuscript entitlewd "Unlocking Agricultural Potential: Analyzing Land Suitability for Key Cereal Crops in

Mansa Watershed, Southwest Ethiopia" Recommended for publica

Reviewer #2: Please make the necessary changes:

Throw some light on the novelty of the study in the introduction section.

Line no. 456-458: Sentences were repeated.

7. PLOS authors have the option to publish the peer review history of their article (what does this mean? ). If published, this will include your full peer review and any attached files.

**Do you want your identity to be public for this peer review?** For information about this choice, including consent withdrawal, please see our Privacy Policy .

Reviewer #1: **Yes: ** Kuldip Jayaswall

Reviewer #2: **Yes: ** Dr. Bindia Gupta

---

## [Author Response · Author response to Decision Letter 2]

13 May 2024

We tried to answer the worries forwared by the reviwers and acad

Question/comments: The title of the paper: Please remove ‘Unlocking Agricultural Potential:”. It doesn’t make sensed.

Answer: Dear reviewer, it revised per your comments: Analyzing Land Suitability for Key Cereal Crops in Mansa Watershed, Southwest Ethiopia in lines 1-2.

Question/comments: The abstract is unclear and the authors better highlight the importance of this research, methods (processing steps), and major findings of the study

Answer: Dear reviewer we tried to amend the comments as indicated in lines 8-21 and 33-34.

Question/comments: Introduction section: AHP and MCDM are not new algorithms. It has been used in many spatial modeling studies. In other words, the specific innovation regarding the methodology should be addressed. What is methodologically research gap the authors want to solve in this study, based on the literature review? Explanations should be added.

• Introduction section, aside from AHP and MCE (Multi-Criteria Evaluation), other methods such as GARP (Genetic Algorithm for Rule-set Prediction), CA-based MCE, and machine-learning based approaches have been commonly applied for land suitability assessment. Please provide advantages and disadvantages of the use of your methods compared to such algorithms.

• Study area and data collection should be presented in separate sections.

Answer: we tried to revise the introduction per the comments indicated in lines 70-122.

Question/comments: The size of the study region is relatively small, while the spatial resolution of geospatial datasets (e.g., rainfall, soil properties) is coarse. Can you extend the study area to a larger region? The soil map used in this study has a scale of 1:250,000. Please consider using the global soil map (250 m resolution), which is offering free of charge about the world.

• What is the rationale for selecting the study area? Describe more details about the study area as well as major crops being analyzed.

Answer: We have been in use because soil maps at the national level provide more precise and in-depth details about the properties of the soil within a given nation or area. Typically, these maps are derived from more regional data sources, like field observations, lab analyses, and soil surveys. Compared to global soil maps, country-level soil maps offer more precision and depth, thereby rendering them more appropriate for application in evaluations of the environment, agricultural planning, as well as targeted land management within that particular nation or region. Conversely, a global overview of soil types and characteristics is typically provided by a world soil map. The accuracy of the information provided may vary as well as may not be reliable due to the large scale and generalization involved in creating world soil maps.

Question/comments:

What is the rationale for selecting the study area? Describe more details about the study area as well as major crops being analyzed.

Answer: Dear reviewer, in the study watershed, such type of study is the first in its kind and the stated problems are observed in the watershed. Moreover, the watershed is selected for academic purpose for such major crops (teff, maize, and wheat). These crops are the major crops used for food security. It is indicated in the main document of the manuscript.

Question/comments: Table 1: change 30m*30m to 30m×30m.

Answer: Look at line 175.

Question/comments: Ground data collection; please mention how you design sampling methods for collecting ground reference samples, and accuracy of the ground truth.

Answer: Literatures and Google earth used and the researcher is acquainted with the watershed to compromise the ground truth with that of Satellite image information or other data sources (kindly see the method section)

Question/comments: ‘Kappa Coefficient’ should be combined with ‘Accuracy Assessment’.

Answer: Dear reviewer, we can suggest the following response that the kappa coefficient is not an index of accuracy; indeed it is not an index of overall agreement but one of agreement beyond chance (Foody, 2020). So Kappa should not be routinely used in accuracy assessment or comparison. So we can omit it from use. Kindly, if you order us to combine the two, we will do it. See the following source: Giles M. Foody. Explaining the unsuitability of the kappa coefficient in the assessment and comparison of the accuracy of thematic maps obtained by image classification. Remote Sensing of Environment, 2020, 239: 111630

Question/comments: In this work, the scores or weights for factors were allocated based on literature and expert opinions. However, the incorporation of expert opinions and results from the literature for suitability analysis reveal limitations, including subjectivity and biases that can introduce uncertainties in the model. Thus, I would suggest that participatory approaches involving multiple experts and local communities or stakeholders to improve the representativeness of collective knowledge while mitigating individual biases to ultimately enhance the overall accuracy of land suitability assessments. Please Questions/ comments: elaborate on your explanations for these concerns in the introduction or methodology section.

Answer: The outputs of the results are confirmed with different stakeholders to improve the representativeness of collective knowledge while mitigating individual biases to ultimately enhance the overall accuracy of land suitability assessments. Look in Lines 70–122 in the main document.

Elaborate classification methods (unsupervised and supervised algorithms) used for mapping LULC types. For example, how did you select LULC classes, training datasets for model training and validation? Detailed descriptions about classification methods must be addressed.

Answer: we tried the following responses as indicated in the lines 242-271.

Questions/ comments: There should be a more thorough discussion of uncertainties and cautions about the application of the methodology in other regions in the discussion section. The comparison to related studies should be addressed.

Answer: Dear Reviewer and editor, the authors tried to address the worries raised, for any queries we’ll ready to do so. For instance, in lines 318-320, 337-339, etc.

Questions/ comments: There are still grammatical issues and inappropriate sentence structures; it is recommended that the authors should hire a professional English editor to go through the editing.

Answer: Respect Dear Reviewer. We tried our best, to correct grammatical errors.

Questions/ comments: Some minor corrections: The quality of all figures should be improved. Tittles of figures can be noted in the caption. For example, Figure: Soil depth suitability: (a) reclassified soil depth for Teff and maize, and (b) reclassified soil depth for wheat.

Answer: Revised in lines 528, 530

Questions/ comments: Math notations in all equations should follow the format of the journal.

Answer: Dear review, We tried but will adjust if there are errors any more.

---

## [Decision Letter · Decision Letter 2]

23 Oct 2024

PONE-D-23-35036R2

Analyzing Land Suitability for Key Cereal Crops in Mansa Watershed, Southwest Ethiopia

PLOS ONE

Dear Dr. g,

Thank you for submitting your manuscript to PLOS ONE. After careful consideration, we feel that it has merit but does not fully meet PLOS ONE’s publication criteria as it currently stands. Therefore, we invite you to submit a revised version of the manuscript that addresses the points raised during the review process.

We look forward to receiving your revised manuscript.

Kind regards,

Nguyen-Thanh Son, Ph.D.

Academic Editor

PLOS ONE

Journal Requirements:

Reviewers' comments:

Reviewer's Responses to Questions

**Comments to the Author**

1. If the authors have adequately addressed your comments raised in a previous round of review and you feel that this manuscript is now acceptable for publication, you may indicate that here to bypass the “Comments to the Author” section, enter your conflict of interest statement in the “Confidential to Editor” section, and submit your "Accept" recommendation.

Reviewer #1: All comments have been addressed

Reviewer #3: All comments have been addressed

2. Is the manuscript technically sound, and do the data support the conclusions?

Reviewer #1: Yes

Reviewer #3: Yes

3. Has the statistical analysis been performed appropriately and rigorously? 

Reviewer #1: Yes

Reviewer #3: Yes

4. Have the authors made all data underlying the findings in their manuscript fully available?

Reviewer #1: Yes

Reviewer #3: Yes

5. Is the manuscript presented in an intelligible fashion and written in standard English?

Reviewer #1: Yes

Reviewer #3: Yes

6. Review Comments to the Author

Reviewer #1: Author mist reduce abstract size. It must be crispy and to the point.

Material and methods must be rewritten (at least one page, discuss in detail)

Conclusion are suggested to improve

Reviewer #3: This research makes a valuable contribution to the field by offering an in-depth analysis of land suitability for major cereal crops in the Mansa Watershed, Southwest Ethiopia. Some suggestions are as follows,

1. The methodology is robust and clearly described, but a brief section explaining why factors like market access or irrigation were not included could make the study more comprehensive. This would help expand the discussion on how future work could integrate such socioeconomic factors, which are vital for agricultural planning.

2. The use of AHP for weighting criteria is appropriate, but the subjectivity in weight assignments could be further addressed. Including a small discussion on how expert opinions were selected and any steps taken to reduce bias would strengthen the methodological transparency and provide confidence in the results.

3. The spatial resolution of some datasets, especially soil and rainfall, may limit the precision of your findings. Including a sentence or two discussing these limitations and justifying why coarser-resolution data were used, or suggesting the potential benefits of using finer-scale datasets in future studies, would enhance the clarity of the analysis.

4. The discussion on uncertainties in the model could be expanded. While the methodology is sound, every model-based analysis carries inherent uncertainties. A brief addition discussing potential sources of uncertainty, such as biases in expert opinions or limitations of the data, and how these may affect generalizability, would add depth to the discussion section.

The paper is methodologically sound and provides clear results; And the authors are encouraged to incorporate these suggestions to refine their analysis and presentation, ensuring that their study reaches its full potential for publication.

7. PLOS authors have the option to publish the peer review history of their article (what does this mean? ). If published, this will include your full peer review and any attached files.

**Do you want your identity to be public for this peer review?** For information about this choice, including consent withdrawal, please see our Privacy Policy .

Reviewer #1: **Yes: ** Kuldip Jayaswall

Reviewer #3: No

---

## [Author Response · Author response to Decision Letter 3]

8 Dec 2024

Journal: PLOS ONE

Manuscript number: PONE-D-23-35036R2

Title: Analyzing Land Suitability for Key Cereal Crops in Mansa Watershed, Southwest Ethiopia

We tried to answer the worries forwarded by the reviewers and academic editor as follows:

Dear editor and reviewers,

Response to Reviewers and editors

We the authors of this manuscript have tried to address the responses to the comments given to the reviewers and editors questions as follows:

Editor request1: Please review your reference list to ensure that it is complete and correct. If you have cited papers that have been retracted, please include the rationale for doing so in the manuscript text, or remove these references and replace them with relevant current references. Any changes to the reference list should be mentioned in the rebuttal letter that accompanies your revised manuscript. If you need to cite a retracted article, indicate the article’s retracted status in the References list and also include a citation and full reference for the retraction notice.

Answer for editor request1: Some of the references are amended in the manuscript for instance, see in references 3, 10, 14, 16, 17, and 18 are replaced and references 68 and 69 are added and other ones are improved in reference section (please see in attached track manuscript).

Reviewer #1 Comment and question::

• Author must reduce abstract size. It must be crispy and to the point. Material and methods must be rewritten (at least one page, discuss in detail)

• Conclusion are suggested to improve

Answer: The abstract has reduced in size and methods and conclusions have rewritten and improved (see attached in track manuscript).

Reviewer #3 comment and question 1: The methodology is robust and clearly described, but a brief section explaining why factors like market access or irrigation were not included could make the study more comprehensive. This would help expand the discussion on how future work could integrate such socioeconomic factors, which are vital for agricultural planning.

Answer for reviewer #3; question 1:

Physical land suitability focuses on the natural characteristics of the land itself, factors like market accessibility and irrigation facilities are part of a broader assessment of land suitability that incorporates human influences and infrastructural considerations. Each type of assessment serves a different purpose in land-use planning and decision-making. Factors like market access or irrigation and other socioeconomic factors which help on future work are integrated and have tried to be discussed in the manuscript which is vital for agricultural planning (see track manuscript in lines 576-578).

Reviewer #3 comment and question 2:

The use of AHP for weighting criteria is appropriate, but the subjectivity in weight assignments could be further addressed. Including a small discussion on how expert opinions were selected and any steps taken to reduce bias would strengthen the methodological transparency and provide confidence in the results.

Answer for reviewer #3: question 2:

Structured decision-making framework, quantifiable results, Expert-driven insights, consistency checking and integration with geographic information systems (GIS) were the necessary steps in reducing of subjectivity bias of AHP. AHP is effective for weighting criteria, enhancing the transparency of the expert selection and weight assignment processes can greatly mitigate the subjectivity inherent in decision-making. Employing systematic strategies for selecting experts, reducing bias through consistency checks and sensitivity analyses, and clearly communicating these steps can strengthen the overall reliability of our results derived through AHP. In addition to these AHP provides a more structured, quantifiable, and reliable methodology for land suitability analysis. It allows for expert input, consistency checking, and effective integration with GIS, resulting in more informed decision-making that can readily adapt to changing priorities and enhance spatial analyses.

Reviewer #3 comment and question 3:

The spatial resolution of some datasets, especially soil and rainfall, may limit the precision of your findings. Including a sentence or two discussing these limitations and justifying why coarser-resolution data were used, or suggesting the potential benefits of using finer-scale datasets in future studies, would enhance the clarity of the analysis.

Answer for reviewer #3: question 3:

The spatial resolution of datasets, particularly for soil and rainfall, can significantly impact the precision of our findings. In this analysis, coarser-resolution data were utilized due to their broader availability and the need for a comprehensive overview at a regional scale. However, the limitations of this approach include potential inaccuracies in localized assessments and variability that finer-scale datasets could capture. Future studies would benefit from integrating higher-resolution datasets, which could provide more precise insights into spatial heterogeneity and improve the reliability of our conclusions. This would allow for a more nuanced understanding of the interactions between soil properties and rainfall patterns, ultimately enhancing the robustness of the research outcomes

Here we attached the revised manuscript with track manuscript changes, clean manuscript, tables in the main manuscript, and all dataset and figure images in separate files.

Best regards

Wakshum Shiferaw (PhD)

Arba Minch University

Phone: +251-911972481

---

## [Editor Report · Decision Letter 3]

23 Dec 2024

PONE-D-23-35036R3Analyzing Land Suitability for Key Cereal Crops in Mansa Watershed, Southwest EthiopiaPLOS ONE

Dear Dr.  g,

Thank you for submitting your manuscript to PLOS ONE. After careful consideration, we feel that it has merit but does not fully meet PLOS ONE’s publication criteria as it currently stands. Therefore, we invite you to submit a revised version of the manuscript that addresses the points raised during the review process.

We look forward to receiving your revised manuscript.

Kind regards,

Nguyen-Thanh Son, Ph.D.

Academic Editor

PLOS ONE

Additional Comments:

Please remove unnecessary names and headings from all maps, such as "reclassified soil depth for Teff and maize; soil depth suitability; WGS_1984_UTM... Mecator" and enhance map captions accordingly. some of references are relatively old. Please update them with the most recent studies.

---

## [Author Response · Author response to Decision Letter 4]

28 Jan 2025

Date 1-27-2025

Wude Taye1, Wakshum Shiferaw1*, Genaye Tsegaye1

1Arba Minch University, College of Agricultural Sciences, Natural Resource Management, P.O.Box-21 Arba Minch, Ethiopia

*Corresponding author: waaqsh@yahoo.com

Dear Reviewers,

Thank you for your valuable feedback on our manuscript. We appreciate the time and effort you have dedicated to reviewing our work. In response to your comments, we have made the following revisions:

1. Map Annotations: We have removed unnecessary names and headings from all maps, including "reclassified soil depth for Teff and maize; soil depth suitability; WGS_1984_UTM... Mecator." We have also enhanced the map captions for clarity and conciseness (Please see figures)

2. References Update: We have reviewed the references and updated any that were relatively old with more recent studies. This ensures our work reflects the latest research in the field. Please see the track manuscript in the revised version of our manuscript

We believe these revisions have strengthened our manuscript and addressed your concerns. Thank you once again for your insightful comments.

Sincerely,

Wakshum Shiferaw (PhD)

Assoc. Prof of Plant Biology and Biodiversity Management

Arba Minch University, Ethiopia

---

## [Editor Report · Decision Letter 4]

27 Feb 2025

PONE-D-23-35036R4Analyzing Land Suitability for Key Cereal Crops in Mansa Watershed, Southwest EthiopiaPLOS ONE

Dear Dr. g,

Thank you for submitting your manuscript to PLOS ONE. After careful consideration, we feel that it has merit but does not fully meet PLOS ONE’s publication criteria as it currently stands. Therefore, we invite you to submit a revised version of the manuscript that addresses the points raised during the review process.

We look forward to receiving your revised manuscript.

Kind regards,

Nguyen-Thanh Son, Ph.D.

Academic Editor

PLOS ONE

Additional Editor Comments:

In the revised version of "Land Suitability Analysis for Sustainable Production of Selected

Cereals in Southeastern Ethiopia", I don't see the land suitability maps. Please make sure that they are presented in the revised manuscript and also discuss about spatial distributions of land suitability classes. Please improve the quality and captions of all figures.

---

## [Author Response · Author response to Decision Letter 5]

4 Mar 2025

Response to Reviewers

Subject: Sending revised PONE-D-23-35036R4

Title: Analyzing Land Suitability for Key Cereal Crops in Mansa Watershed, Southwest Ethiopia

Journal: PLOS ONE

Editor Comments:

Questions: I don't see the land suitability maps.

Dear Prof. Nguyen-Thanh Son Academic Editor,

Response: We tried to forward the suitability maps with the manuscript and in separate files. W added also the last figure 14 if the manuscript makes clear even though the license issue of the figure requested by an editor of the journal.

Questions: Please make sure that they are presented in the revised manuscript and also discuss about spatial distributions of land suitability classes.

Response: The authors tried revise and discuss spatial distributions of land suitability classes, for instance, look at lines 305-306, 325-327, 342-345, 365-368, 383-386, 403-405, 428-431, 443-445, 470-471, 504-506.

Questions: Please improve the quality and captions of all figures.

Response: Thank you we tried to improve the quality of the figures and figure captions. See in the track manuscript.

---

## [Editor Report · Decision Letter 5]

3 Jul 2025

Analyzing Land Suitability for Key Cereal Crops in Mansa Watershed, Southwest Ethiopia

PONE-D-23-35036R5

Dear Dr. Wakshum Shiferaw,

We’re pleased to inform you that your manuscript has been judged scientifically suitable for publication and will be formally accepted for publication once it meets all outstanding technical requirements.

Kind regards,

Mario Licata, Ph.D.

Academic Editor

PLOS ONE

Additional Editor Comments (optional):

Dear authors,

I am satisfied by the work the authors have done. They have followed all the reviewer's recommendations and improved the manuscript. No gaps and issued at this stage. No further comments to authors from my side.
---

## [Editor Report · Acceptance letter]

PONE-D-23-35036R5

PLOS ONE

Dear Dr. Shiferaw,

I'm pleased to inform you that your manuscript has been deemed suitable for publication in PLOS ONE. Congratulations! Your manuscript is now being handed over to our production team.

Kind regards,

on behalf of

Dr. Mario Licata

Academic Editor

PLOS ONE